# A-PSRO: A Unified Strategy Learning Method with Advantage Metric for Normal-form Games

Yudong Hu [1]   Haoran Li [1]   Congying Han [1]   Tiande Guo [1]   Mingqiang Li [2]   Bonan Li [1]

## Abstract

Solving the Nash equilibrium in normal-form games with large-scale strategy spaces presents significant challenges. Open-ended learning frameworks, such as PSRO and its variants, have emerged as effective solutions. However, these methods often lack an efficient metric for evaluating strategy improvement, which limits their effectiveness in approximating equilibria. In this paper, we introduce a novel evaluative metric called Advantage, which possesses desirable properties inherently connected to the Nash equilibrium, ensuring that each strategy update approaches equilibrium. Building upon this, we propose the Advantage Policy Space Response Oracle (A-PSRO), an innovative unified open-ended learning framework applicable to both zero-sum and general-sum games. A-PSRO leverages the Advantage as a refined evaluation metric, leading to a consistent learning objective for agents in normal-form games. Experiments showcase that A-PSRO significantly reduces exploitability in zero-sum games and improves rewards in general-sum games, outperforming existing algorithms and validating its practical effectiveness.

## 1. Introduction

The Nash equilibrium in normal-form games, encompassing both zero-sum and general-sum scenarios, is a fundamental concept for modeling the behavior of rational, utility-maximizing agents. By approximating these equilibria, agents that outperform humans have been developed in various domains, including chess (Silver et al., 2018), poker (Brown & Sandholm, 2019), and real-time strategy (RTS) games (Vinyals et al., 2019; Berner et al., 2019). However, achieving equilibrium becomes increasingly challenging in large-scale games with complex strategy spaces (Hernandez-Leal et al., 2017). The development of a unified and efficient equilibrium solver remains a challenge.

The Policy Space Response Oracle (PSRO) offers an efficient open-ended strategy learning framework (Lanctot et al., 2017). Due to its scalability, numerous subsequent works have developed various PSRO variants to enhance the efficiency of solving specific games. In zero-sum games, strategies that emphasize diversity have proven effective for learning Nash equilibrium (Balduzzi et al., 2019). Methods such as UDF-PSRO (Liu et al., 2021), UDM-PSRO (Liu et al., 2022) and PSD-PSRO (Yao et al., 2023), which are based on diversity modeling, are among the most efficient equilibrium solvers for large-scale zero-sum games. However, while increasing diversity improves exploration efficiency, it often results in inefficient strategy improvement due to the lack of proper guidance. To address this issue, our work introduces a novel approach by integrating the concept of Advantage as an independent evaluative metric for strategies. We establish a mathematical equivalence between advantage maximization and Nash equilibrium, thereby supporting the use of advantage as a strategic learning objective within the PSRO framework.

In general-sum games, Nash equilibria consist of multiple joint strategies, each associated with different rewards (Foerster et al., 2018). This contrasts with symmetric zero-sum games, where the Minimax property ensures that all Nash equilibria yield identical rewards (Li et al., 2019). Previous efforts to improve the PSRO algorithm in general-sum games have primarily focused on enhancing the efficiency of equilibrium solving (Zhang et al., 2021), often neglecting the differences in rewards between equilibria. However, recent research has highlighted that different objectives in strategy learning can lead to distinct equilibria (Willi et al., 2022). By adopting an appropriate objective, agents can achieve equilibrium strategies that also maximize rewards (Hu et al., 2022). In this work, we present a method to enhance strategy rewards using the advantage function. This approach enables our algorithm to achieve higher rewards compared to other PSRO methods.

[1] University of Chinese Academy of Sciences, Beijing, China [2] Information Science Academy, China Electronics Technology Group Corporation, Beijing, China. Correspondence to: Congying Han < hancy@ucas.ac.cn >.

*Proceedings of the $42^{nd}$ International Conference on Machine Learning*, Vancouver, Canada. PMLR 267, 2025. Copyright 2025 by the author(s).

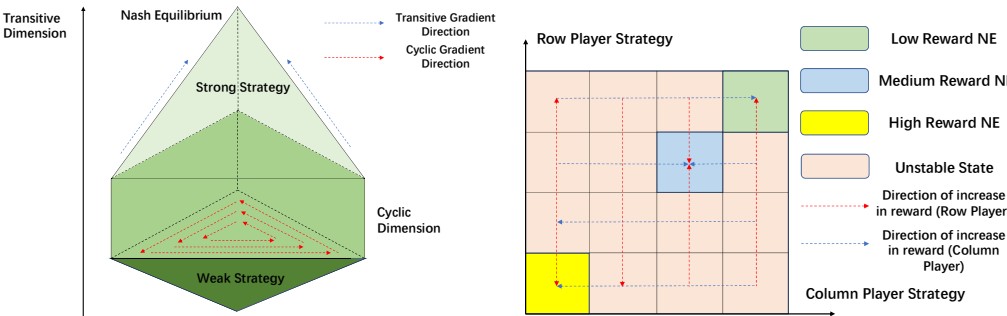

(a) Zero-sum game geometrical structure        (b) General-sum game geometrical structure

*Figure 1.* The geometrical structure examples of zero-sum games and general-sum games. Figure (a) shows the structure of a zero-sum game with both transitive and cyclic dimensions. The direction of the strategy gradient refers to the expected updates for a strategy that maximizes the reward. Figure (b) shows the structure of a general-sum game with multiple equilibria. The independent learning process of the agents leads to the update of the strategy in the direction indicated by the arrow.

In summary, we introduce A-PSRO, an improved equilibrium solver for large-scale normal-form games that leverages the Advantage function. In symmetric zero-sum games, the Advantage function exhibits favorable properties, enabling agents to approach the Nash equilibrium deterministically. By incorporating the Advantage function into existing diversity-based approaches, we achieve significant improvements in learning Nash equilibrium strategies. In general-sum games, although the Advantage function is non-convex, its local maxima correspond to equilibria with varying rewards. By exploring strategies near the global optimum with the objective of maximizing the Advantage function, our algorithm converges to equilibria with higher rewards. We also provide methods for both exact and approximate computation of the Advantage function, allowing A-PSRO to integrate seamlessly into existing PSRO frameworks.

We conducted experiments on various games to evaluate our algorithm. In zero-sum games, the strategies derived using the A-PSRO algorithm are significantly closer to equilibrium. In general-sum games, the A-PSRO algorithm enables agents to learn strategies that achieve globally optimal rewards, avoiding entrapment in locally optimal equilibria. These results underscore the effectiveness of the A-PSRO algorithm as a unified framework for solving equilibrium.

## 2. Related Work and Background

In this paper, we focus on normal-form games with finite dimensions, typically represented by three key elements denoted as $(\mathcal{N}, \mathcal{A}, \mathcal{U})$. Here, $\mathcal{N}$ represents the players in the game, $\mathcal{A}$ denotes the action (pure strategy) space of the players, and $\mathcal{U}$ refers to the utility function. In such games, agents generally adopt strategies $\pi$ rather than directly choosing actions $a \in \mathcal{A}$. $\pi$ is defined as a probability distribution over actions: $\pi = (p_1, p_2, \cdots, p_{|\mathcal{A}|})$, $\sum p_i = 1$,

where $p_i$ represents the probability of choosing action $a_i$. To clearly distinguish between different strategies, we use $\pi_i^t$ to denote the $t$-th strategy of player $i$.

The Nash equilibrium (NE) characterizes a stable state, where no agent can increase its reward by unilaterally altering its strategy. For the joint strategy $(\pi_1, \cdots, \pi_n)$, it is an NE when $\forall i \in \{1, \cdots, n\}$, $\pi_i$ is the best response (BR) to the strategies of other agents: $\forall \pi_i'$, $U_i(\pi_i', \pi_{-i}) \leq U_i(\pi_i, \pi_{-i})$ ($U_i$ denotes the expected reward of agent $i$, $\pi_{-i}$ represents the joint strategy except for agent $i$).

Exploitability is defined as the distance of joint strategy $(\pi_i, \pi_{-i})$ and the Nash Equilibrium:

$$\mathcal{E}(\pi_i, \pi_{-i}) = \sum_{k=1}^{n} [\max_{\pi_k^*} U_k(\pi_k^*, \pi_{-k}) - U_k(\pi_k, \pi_{-k})].$$

If the exploitability of a joint strategy $(\pi_i, \pi_{-i})$ is 0, it is a Nash equilibrium.

### 2.1. Symmetric Zero-sum Games with Transitive Dimension and Cyclic Dimension

Symmetric zero-sum games with two players $(i, j)$ are among the most studied game forms because their model is consistent with many real-world scenarios (Zhang & Sandholm, 2022; Sokota et al., 2023). In such games, the joint strategy of the agents is $(\pi_i, \pi_j)$, with their rewards defined as $U_i(\pi_i, \pi_j) = -U_j(\pi_i, \pi_j)$. The symmetric property implies that both agents share the same strategy space $\Pi$, and it holds that $U_i(\pi_i^1, \pi_j^2) = U_j(\pi_i^2, \pi_j^1)$.

Previous studies have shown that the geometric structure of symmetric zero-sum games resembles a spinning top, consisting of both transitive and cyclic dimensions (illustrated in Figure 1(a)) (Czarnecki et al., 2020). The transitive dimension characterizes the absolute strengths between strategies.

A game is considered transitive if there exists an evaluation function for the strength of a strategy, denoted as $f_v(\pi)$. In strategic interactions, the strategy with a higher evaluation function always yields a higher reward.

$$U_i(\pi_i^1, \pi_j^2) = f_v(\pi_i^1) - f_v(\pi_j^2) = -U_j(\pi_i^1, \pi_j^2).$$

The cyclic dimension indicates the presence of mutual restraint among strategies, similar to the dynamics observed in Rock-Scissor-Paper (RSP). In a game with only cyclic dimension, for any strategy $\pi_i^1$ in the strategy space $\Pi$, its expectation of reward when facing other strategies is 0:

$$\int_{\pi_j^0 \in \Pi} U_i(\pi_i^1, \pi_j^0) \cdot d\pi_j^0 = 0.$$

Real-world games typically exhibit both transitive and cyclic dimensions, making it impracticable to evaluate strategies directly using the aforementioned equations. Strategy updates based on gradients may become trapped in the cyclic dimensions, leading to slow convergence towards the Nash equilibrium in the transitive dimensions.

### 2.2. General-sum Games with Multiple Equilibria

Unlike zero-sum games, general-sum games typically feature multiple Nash equilibria with varying rewards (Tang et al., 2021), as illustrated in Figure 1(b). Previous studies have often focused on specific types of equilibria, such as Pareto-optimal equilibria (Sen et al., 2003), MENE (Maximum Entropy Nash Equilibrium) (Balduzzi et al., 2018), and optimal PSNE (Pure Strategy Nash Equilibrium) (Nguyen et al., 2023).

In the theory of learning in games, the update rule for each iteration determines the learned equilibrium. Recent studies have focused on improving equilibrium rewards while ensuring convergence to an equilibrium (Foerster et al., 2018; Letcher et al., 2019). It has been demonstrated that providing appropriate guidance to agents can effectively enhance the utility of the learning process (Hu et al., 2022).

### 2.3. Open-ended Learning Framework

Prior research has introduced various methods for solving Nash equilibria, including WOLF (Win or Learn Fast) (Bowling & Veloso, 2001) and AWESOME (Adapt When Everybody is Stationary, Otherwise Move to Equilibrium) (Conitzer & Sandholm, 2003). The most widely used algorithm is fictitious play, favored for its simplicity of execution (Fudenberg & Levine, 1995). There are also algorithms based on adaptive game playing (Freund & Schapire, 1999). It is worth noting that due to the complexity of general-sum game structures, these algorithms typically exhibit guaranteed convergence only in zero-sum games.

The PSRO algorithm presents an effective approach to solving Nash equilibrium in games with large-scale strategy spaces. Inspired by the Double Oracle algorithm (McMahan et al., 2003; Bošanský et al., 2016), PSRO establishes a population to represent strategies for each agent. The initial strategy population is generated randomly as $\mathcal{P}_i = (\pi_i^1, \cdots, \pi_i^t)$. In each iteration, the empirical game matrix for agent $i$ is calculated as $\mathcal{M}_i = \mathcal{P}_i \times U_i \times \mathcal{P}_{-i}$. By adopting the fictitious play to solve the Nash equilibrium of meta-game, with a payment matrix of $(\mathcal{M}_i, \mathcal{M}_{-i})$, we can derive the meta-equilibrium for agents: $(\theta_i, \theta_{-i})$. Then, agent $i$ will search for a new strategy $\pi_i^{t+1}$, usually the best response to the meta-equilibrium of the opponent $\mathrm{BR}(\theta_{-i})$.

Improvements to the PSRO algorithm primarily involve incorporating new meta-game solvers or adopting diverse objectives to guide the generation of new strategies. Additionally, there are open-ended algorithms that explore alternative equilibrium concepts, such as the $\alpha$-Rank equilibrium ($\alpha$-PSRO) (Muller et al., 2020), the correlated equilibrium (JPSRO) (Marris et al., 2021) and coarse correlated equilibrium (CCE) (Liu et al., 2024).

In this paper, we focus on refining the process of strategy exploration of PSRO framework. Previous works have typically enhanced strategy generation by increasing diversity. Several methods exist for measuring diversity, including Expected Cardinality (EC) (Perez-Nieves et al., 2021), Behavioral Diversity (BD), Response Diversity (RD) (Liu et al., 2021) and Policy Space Diversity (PSD) (Yao et al., 2023).

## 3. From Exploitability to Advantage Function

We first consider symmetric zero-sum games, where both agents share the same strategy space $\Pi = \{\pi^1, \pi^2, \cdots\}$. From the symmetry, we have the following property:

**Theorem 3.1.** *In symmetric zero-sum games, if the joint strategy $(\pi_i^1, \pi_j^2)$ is a Nash equilibrium, we have $(\pi_i^1, \pi_j^1)$ and $(\pi_i^2, \pi_j^2)$ are both Nash equilibriums.*

For a strategy $\pi$ in a zero-sum game, its best response $\mathrm{BR}(\pi)$ is usually a set containing many strategies. However, we have the following property:

**Theorem 3.2.** *For any two-player game, when the strategy of another player is fixed (denoted as $\pi_j$), there always exists pure strategy $a_i \in \mathcal{A}$ which satisfies that $a_i \in \mathrm{BR}(\pi_j)$. Particularly, in zero-sum games, $U_i(\pi_i, \pi_j)$ is always the same for all $\pi_j \in \mathrm{BR}(\pi_i)$.*

Then we have:

$$\begin{aligned}
\mathcal{E}(\pi_i^1, \pi_j^2) &= \max_{\pi_{i'}} U_i(\pi_{i'}^1, \pi_j^2) - U_i(\pi_i^1, \pi_j^2) \\
&\quad + \max_{\pi_{j''}} U_j(\pi_i^1, \pi_{j''}^2) - U_j(\pi_i^1, \pi_j^2) \\
&= U_i(\mathrm{BR}(\pi_j^2) \cap \mathcal{A}, \pi_j^2) + U_j(\pi_i^1, \mathrm{BR}(\pi_i^1) \cap \mathcal{A}) \\
&= -U_j(\mathrm{BR}(\pi_j^2) \cap \mathcal{A}, \pi_j^2) - U_i(\pi_i^1, \mathrm{BR}(\pi_i^1) \cap \mathcal{A}).
\end{aligned}$$

According to Theorem 3.1, the symmetry in most 2p0s games allows us to define the distance between a single strategy and equilibrium, similar to the exploitability. If we consider $\mathrm{BR}(\pi_i^1)$ as a function of $\pi_i^1$, then the value of $U_i(\pi_i^1, \mathrm{BR}(\pi_i^1) \cap \mathcal{A})$ is determined only by $\pi_i^1$.

**Definition 3.3.** In two-player zero-sum games, we define the advantage function as:

$$\mathcal{V}_i(\pi_i) = U_i(\pi_i, a_j^0), \ a_j^0 \in \mathrm{BR}(\pi_i) \cap \mathcal{A}_j.$$

From Theorem 3.2, we can see that this definition makes sense because the selection of best response $\mathrm{BR}(\pi_i)$ does not affect the value of $\mathcal{V}_i$. Therefore, we also use $\mathcal{V}_i(\pi_i) = U_i(\pi_i, \mathrm{BR}(\pi_i) \cap \mathcal{A}_j)$ to denote the advantage function.

**Theorem 3.4.** *In two-player zero-sum games,*

- $\mathcal{E}(\pi_i, \pi_j) = -(\mathcal{V}_i(\pi_i) + \mathcal{V}_j(\pi_j))$.

- $\mathcal{V}_i(\pi_i)$ *is Lipschitz continuous about* $\pi_i$*, and* $-\mathcal{V}_i(\pi_i)$ *is a convex function about* $\pi_i$*.*

- *If the game is symmetric,* $\forall \pi_i$*,* $\mathcal{V}_i(\pi_i) \leq 0$*. The joint strategy* $(\pi_i^0, \pi_j^0)$ *is a Nash equilibrium if and only if* $\mathcal{V}_i(\pi_i^0) = \mathcal{V}_j(\pi_j^0) = 0$*. In games with only transitive dimension,* $\mathcal{V}_i(\pi_i) > \mathcal{V}_j(\pi_j)$ *implies* $U_i(\pi_i, \pi_j) > 0$*.*

From Theorem 3.4, we observe that the advantage function has a unique local and global maximum of 0 in symmetric zero-sum games, indicating that the corresponding strategy is a Nash equilibrium. This finding suggests that improving the advantage of strategies guides the learning process towards convergence at the Nash equilibrium. Additionally, the advantage function can be computed within the pure strategy space $\mathcal{A}$.

## 4. Advantage Policy Space Response Oracle

In this section, we introduce the A-PSRO framework and its theoretical properties for zero-sum and general-sum games.

### 4.1. A-PSRO for Symmetric Zero-sum Games

In classic PSRO algorithms, new strategies added to the population $\Pi_i = \{\pi_i^1, \cdots, \pi_i^t\}$ are typically derived through best response, with the opponent strategy fixed as the meta-Nash strategy:

$$\pi_i^{t+1} \in \mathrm{BR}(\theta_j), \text{ where } (\theta_i, \theta_j) = \mathrm{NE}(\mathcal{M}_i, \mathcal{M}_j).$$

Best response-based updates may stagnate within cyclic dimensions, causing the PSRO algorithm to converge slowly in non-transitive games. To address this, diversity strategy algorithms offer an improvement over the classic PSRO by increasing the probability of discovering novel strategies in

the transitive dimension. For example, DPP-PSRO (Perez-Nieves et al., 2021) incorporates Expected Cardinality (EC) as a regularizer to generate new strategies. However, new strategies generated by diversity algorithms are stochastic, which means they cannot deterministically approach equilibrium.

From Theorem 3.4, we can see that increasing the advantage of strategy will approach the Nash equilibrium. Since $-\mathcal{V}_i(\pi_i)$ is convex, we design the A-PSRO to introduce advantage as the objective of strategy learning.

For the population-based strategy update approach in PSRO, we define $\mathcal{V}_i(\pi_i \mid \mathcal{P}_j) = U_i(\pi_i, \mathrm{BR}(\pi_i \mid \mathcal{P}_j))$, where

$$\mathrm{BR}(\pi_i \mid \mathcal{P}_j) = \mathrm{argmax}_{\pi_j \in \mathcal{P}_j} U_j(\pi_i, \pi_j).$$

**Theorem 4.1.** *In symmetric zero-sum games, given the population* $\mathcal{P}_i = \mathcal{P}_j = \{\pi_i^1, \cdots, \pi_i^t\}$*,* $\forall \pi_i^k \in \mathcal{P}_i$*, we have* $\mathcal{V}_i(\pi_i^k) \leq \mathcal{V}_i(\theta_i \mid \mathcal{P}_i)$*. Here,* $\theta_i$ *is the equilibrium of the meta-game corresponding to the population* $\mathcal{P}_i$*.*

Note that $\mathcal{V}_i(\pi_i^k) \leq \mathcal{V}_i(\theta_i)$ does not always hold (example given in Supplementary Material). However, we have:

$$\forall \pi_i^k \in \mathcal{P}_i, \ \mathcal{V}_i(\pi_i^k) \leq \mathcal{V}_i(\theta_i), \text{when } \mathrm{hull}(\mathcal{P}_i) = \Pi,$$

where $\mathrm{hull}(\mathcal{P}_i)$ is the convex hull of population. Theorem 4.1 indicates that the equilibrium of the meta-game approximately maximizes the advantage of the current population.

We aim to search for a new strategy with a deterministic increase in the advantage of population. We have the following property of the advantage in population iterations.

**Theorem 4.2.** *Given the meta-equilibrium strategy* $\theta_i$*, if* $\mathcal{V}_i(\theta_i) < 0$*, there exists* $\Delta \pi_i \in \mathcal{A}$ *and* $\delta > 0$ *satisfying:*

$$\forall \ 0 < d < \delta, \quad \mathcal{V}_i((1-d) \cdot \theta_i + d \cdot \Delta \pi_i) > \mathcal{V}_i(\theta_i).$$

Theorem 4.2 indicates that if the meta-equilibrium $\theta_i$ of the current population is not a Nash equilibrium, we can find a strategy closer to the Nash equilibrium in its neighborhood. Furthermore, according to optimization theory, since the advantage function is convex, this strategy update guarantees to approach the Nash equilibrium at a sublinear rate.

Next, we explain how to improve PSRO's strategy exploration process by using the advantage function as a regularization term. A-PSRO differs from other algorithms only when adding new strategies $\pi_i^{t+1}$ to the current population $\mathcal{P}_i$, and we refer to this process as LA (LookAhead). Given step size $d$, the new strategy generated is:

$$\pi_i^{t+1} = (1-d) \cdot \theta_i + d \cdot \Delta \pi_i,$$
$$\Delta \pi_i = \mathrm{argmax}_{\Delta \pi \in \mathcal{A}} \mathcal{V}_i((1-d) \cdot \theta_i + d \cdot \Delta \pi).$$

For finite-dimensional zero-sum games, the advantage function can be computed through matrix multiplication:

$$\mathcal{V}_i(\pi_i) = \mathrm{Min}_{a_j \in \mathcal{A}_j} \ \pi_i \times U_i \times a_j,$$

making its computational complexity comparable to that of the best response.

## 4.2. A-PSRO for Two-player General-sum Games

To the best of our knowledge, ensuring convergence in general-sum games remains a challenging problem. Most algorithms can only converge within specific game structures. However, since the advantage function is directly tied to the reward function, it can guide agents to learn strategies that achieve higher rewards. When multiple equilibria are learnable within a game, A-PSRO outperforms general algorithms by identifying equilibria with higher joint rewards.

In a two-player general-sum game, an action $a_i^1$ is called dominated if $\forall \pi_j \in \Pi_j$, there exists $a_i^k \in \mathcal{A}_i \setminus \{a_i^1\}$, $U_i(a_i^1, \pi_j) \leq U_i(a_i^k, \pi_j)$. It is usually assumed that dominated actions can be removed from the game. Therefore, we define the simplified game:

**Definition 4.3.** We call a game a simplified game if there does not exist dominated pure strategy $a_i$ for any player $i$.

For arbitrary general-sum game, a corresponding simplified game can be obtained by removing dominated actions.

When extending Definition 3.3 to general-sum games, different choices of $\mathrm{BR}(\pi_i)$ will lead to inconsistent advantage functions. Fortunately, in simplified games, we have the following property:

**Theorem 4.4.** In two-player simplified games, $\forall \pi_i$, for any $a_j^l \in \mathrm{BR}(\pi_i) \cap \mathcal{A}_j$ and $\forall \delta > 0$, there always exists $\pi_i'$ which satisfies $|\pi_i' - \pi_i| < \delta$ and $\mathrm{BR}(\pi_i') \cap \mathcal{A}_j = \{a_j^l\}$.

This theorem shows that for almost all strategies, their advantage can be defined through a unique best response. To maintain consistency, when multiple best responses from opponents exist, we select the one that maximizes the agent's own reward to define the advantage function:

**Definition 4.5.** In two-player simplified games, we define $\mathcal{V}_i(\pi_i) = \max_{a_j} U_i(\pi_i, a_j)$, where $a_j \in \mathrm{BR}(\pi_i) \cap \mathcal{A}_j$.

This definition always makes sense. Similar to zero-sum games, the advantage function has the following properties in simplified general-sum games:

**Theorem 4.6.** In two-player simplified games,

- $\forall i$, $\mathcal{V}_i(\pi_i)$ is Lipschitz continuous.

- We assume that the joint strategy $(\pi_i, \pi_j)$ is a Nash equilibrium. If $\mathrm{BR}(\pi_i) \cap \mathcal{A}_j$ has the unique element, then $\mathcal{V}_i(\pi_i)$ is a local maximum.

- Under the same assumption, if $(\pi_i^1, \pi_j^2)$ and $(\pi_i^3, \pi_j^4)$ are both NEs, then $(\pi_i^1, \pi_j^2)$ Pareto dominates $(\pi_i^3, \pi_j^4)$ if and only if $\mathcal{V}_i(\pi_i^1) \geq \mathcal{V}_i(\pi_i^3)$ and $\mathcal{V}_j(\pi_j^2) \geq \mathcal{V}_j(\pi_j^4)$.

In general-sum games, the advantage function is non-convex, which means that strategy gradient algorithms only converge to local maxima. However, when computing the meta-equilibrium $(\theta_i, \theta_j)$ within the population-based PSRO algorithm, we prove that there exists a space in which strategy converges to the global optimum.

**Theorem 4.7.** In two-player simplified games, the current population for agent $i$ is $\mathcal{P}_i = \{\pi_i^1, \cdots, \pi_i^t\}$. $\theta_i$ is the global maximum point of the advantege $\mathcal{V}_i$ in $\mathrm{hull}(\mathcal{P}_i)$. Then there must exist a non-zero measure set $\mathcal{D}' \subset \mathrm{hull}(\mathcal{P}_i)$, which satisfies that if $\theta_i'$ is a local maximum point of the advantege $\mathcal{V}_i$ in $\mathcal{D}'$, then $\mathcal{V}_i(\theta_i') = \mathcal{V}_i(\theta_i)$.

Theorems 4.6 and 4.7 establish that in general-sum games, there exists a non-zero measure set near the optimal equilibrium where the population of strategies converges towards that equilibrium. Strategies close to equilibria with optimal rewards tend to have higher advantage values due to the Lipschitz continuity of the advantage function.

In A-PSRO, we adopt a strategy exploration approach designed to increase the probability of discovering strategy with higher advantage:

$$\pi_i^{t+1} = (1 - d) \cdot \theta_i + d \cdot \Delta \pi_i,$$

$$\theta_i = \operatorname*{argmax}_{(\theta_i', \theta_j') \in \Theta} \mathcal{V}_i(\theta_i'), \ \Theta = \bigcup_{\pi_{i,j} \in \mathrm{hull}(\mathcal{P}_{i,j})} \mathcal{O}(\mathcal{P}_i, \mathcal{P}_j \mid \pi_{i,j}).$$

Here, $d$ is the fixed step size, and the calculation of $\Delta \pi_i$ is the same as zero-sum games. $\mathcal{O}(\mathcal{P}_i, \mathcal{P}_j \mid \pi_{i,j})$ represents the meta-equilibrium obtained through a fictitious play oracle with $(\pi_i, \pi_j)$ as initial strategies.

## 4.3. A-PSRO for Large-scale Games

The PSRO framework has also been widely adopted for solving large-scale extensive-form games due to its compatibility with neural network-based implementations. Since this paper primarily focuses on the theoretical properties of the advantage function and its applications in normal-form games, related studies will be presented in future work. Regarding the extension of A-PSRO to other scenarios such as sequential decision-making scenarios, we have conducted some theoretical analysis and experiments. Here, we present several feasible extensions.

**Empirical Games Solver.** The research in (Czarnecki et al., 2020) indicates that pure strategies with a wide range of skills extracted from large-scale extensive-form games (such as StarCraft) can also define a normal-form game. The strategies obtained by solving the empirical game are also important for many problems. There are several works about empirical games such as (Walsh et al., 2004). The A-PSRO algorithm presented in this paper can be directly applied to efficiently solve the aforementioned empirical

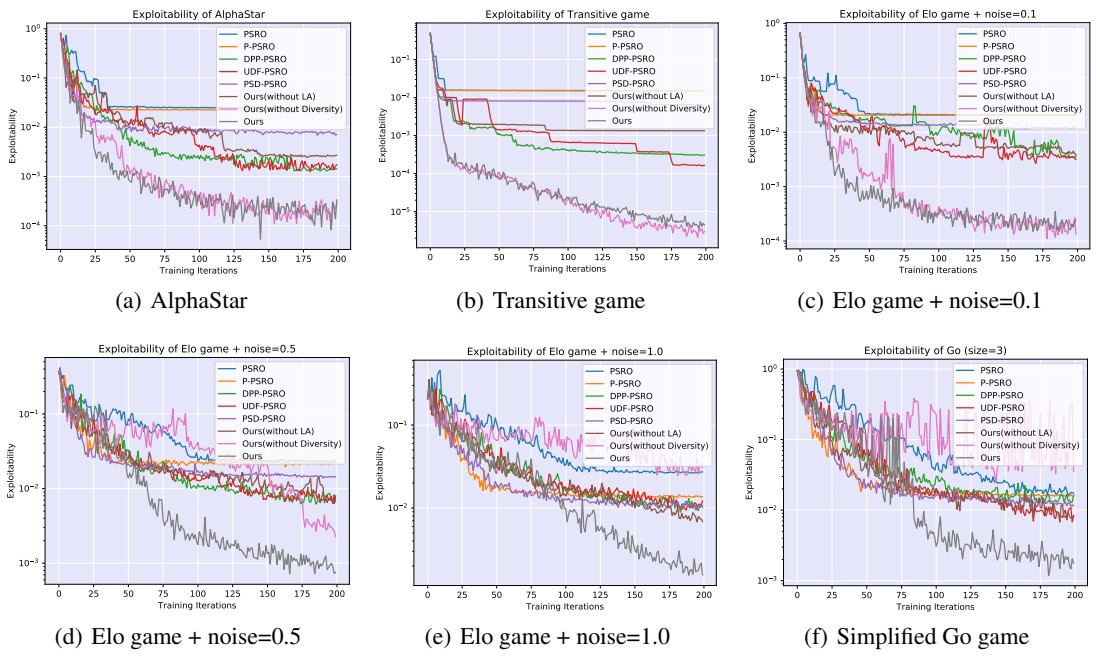

*Figure 2.* The exploitability of the joint strategy learned by agents in various zero-sum games is depicted. The reduction in exploitability through population iterations can serve as an indicator of the effectiveness in approximating the Nash equilibrium.

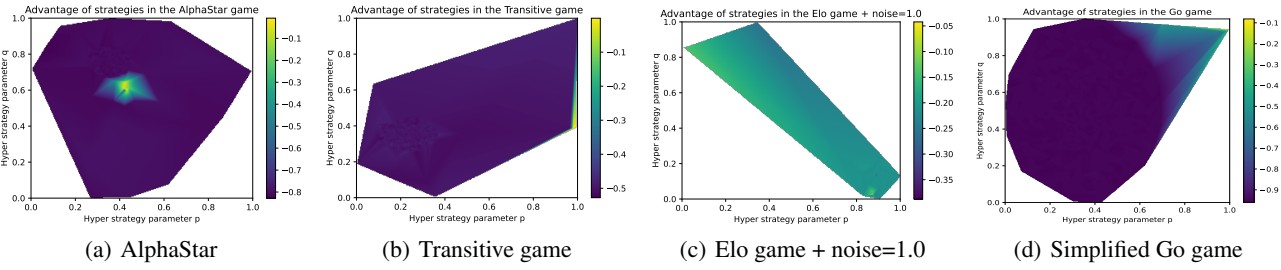

*Figure 3.* The advantage distribution of strategies. Lighter colored regions indicate strategies with higher advantage.

games. The solutions can serve as effective approximations of Nash equilibria for large-scale extensive-form games.

**Neural Network Approximation.** When PSRO is applied to solving large-scale extensive-form games, it typically employs RL methods such as policy gradient to explore new strategies. To introduce regularization based on the advantage function, we need to design neural network approximations of the advantage function.

We sample a set of strategies based on the empirical game as a predefined best response set $\mathcal{A}$. Then we pre-train a best response predictor $\mathrm{BR}^*(\cdot, \gamma)$ with parameter $\gamma$, which can be implemented analogously to a classification task. For any $\pi_i$, we generate the label as the first pure strategy maximizing the reward of $\pi_i$ in its best response set:

$$\mathrm{BR}^{**}(\pi_i) = (c_1, \cdots, c_{l-1}, c_l, c_{l+1}, \cdots, c_{|\mathcal{A}_j|}).$$

$$c_l = 1 \text{ if } l = \mathrm{argmin}_l \left\{ \mathrm{argmax}_{a_j^l \in \mathcal{A}_j} U_j(\pi_i, a_j^l) \right\}.$$

After training the predictor parameter $\gamma$ with cross entropy loss function $-\mathrm{BR}^{**}(\pi_i) \cdot log \, \mathrm{BR}^*(\pi_i, \gamma)$, the approximated advantage function $V_i^*(\pi_i)$ can be computed for each strategy $\pi_i$ using the predicted opponent's best response $\mathrm{BR}^*(\pi_i, \gamma)$. This allows to train the advantage predictor $\hat{\mathcal{V}}_i(\pi_i, \gamma')$ with MSE loss. Base on the advantage predictor, we can calculate $\nabla_{\pi_i} \hat{\mathcal{V}}_i(\pi_i, \gamma')$ with fixed $\gamma'$ to generate the advantage regularization for strategy updating. We demonstrate through the following theorem that when the approximation satisfies certain conditions, it can still effectively guarantee the global effectiveness of strategy exploration.

**Theorem 4.8.** *If* $|\nabla_{\pi_i} \hat{\mathcal{V}}_i(\pi_i) - \nabla_{\pi_i} \mathcal{V}_i(\pi_i)| \leq \frac{1}{3} |\nabla_{\pi_i} \mathcal{V}_i(\pi_i)|$, *the strategy generated by the A-PSRO exploration process will converge to equilibrium strategy with the sublinear convergence rate in symmetric zero-sum games.*

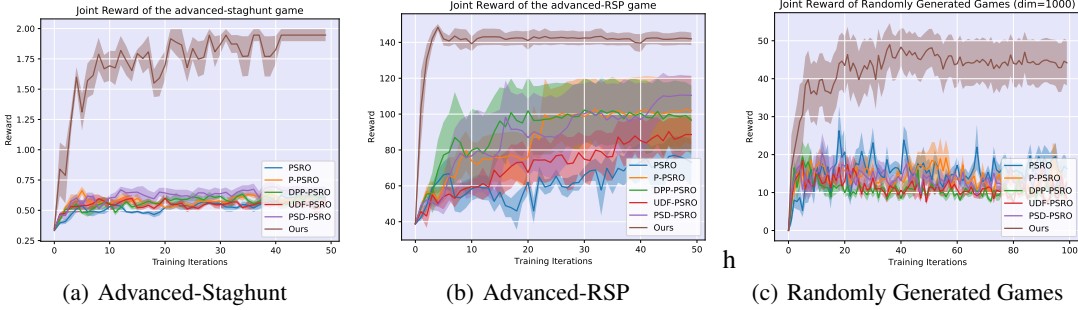

(a) Advanced-Staghunt      (b) Advanced-RSP      (c) Randomly Generated Games

*Figure 4.* The joint reward of the agent system in general-sum games. The Staghunt game and the RSP game are repeated 10 times and averaged for plotting. Randomly generated games contain 100 games with the same reward distribution.

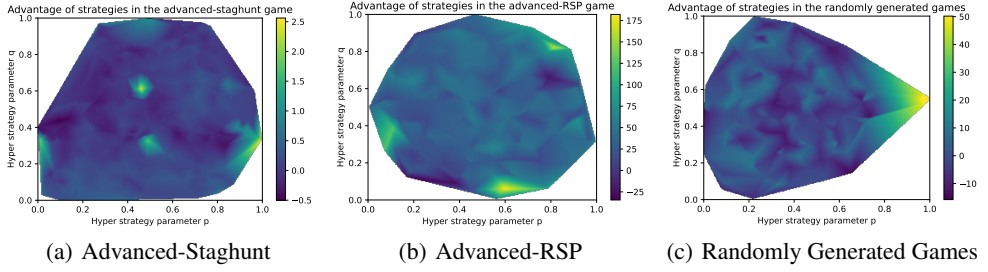

(a) Advanced-Staghunt      (b) Advanced-RSP      (c) Randomly Generated Games

*Figure 5.* The advantage distribution of strategies in different two-player general-sum games.

**Expansion in Multi-player Games.** In $n$-player games ($n > 2$), the advantage function cannot be defined by the best response. We define the advantage function as:

$$\mathcal{V}_i(\pi_i) = \max U_i(\pi_i, \mu(\pi_i)), \ \mu(\pi_i) = \mathcal{O}(\Pi_{-i} \mid \pi_i).$$

$\mu(\pi_i)$ is a joint strategy without player $i$ as the equilibrium of the $(n-1)$-player subgame when the strategy of player $i$ is $\pi_i$. $\mu(\pi_i)$ is computed by an equilibrium oracle $\mathcal{O}$.

In order to efficiently approximate the advantage function, we define the optimistic equilibrium oracle similar to (Basilico et al., 2020; Wang et al., 2022).

$$\mu(\pi_i) = \text{argmax}_{\pi_{-i}} U_i(\pi_i, \pi_{-i}), \ \pi_{-i} = \text{NE}(U_{-i} \mid \pi_i).$$

Here, $\text{NE}(U_{-i} \mid \pi_i)$ represents a Nash equilibrium of the subgame with $\pi_i$ fixed. We give an approximation of the optimistic equilibrium oracle by simplifying it to a two-player game. We view the other agents as a single agent $\{-i\}$ with action space $\mathcal{A}_{-i}$, and approximate advantage as

$$\hat{\mathcal{V}}_i(\pi_i) = \max_{\pi_{-i}} U_i(\pi_i, \pi_{-i}), \ \pi_{-i} \in \text{BR}(\pi_i) \cap \mathcal{A}_{-i}.$$

Based on the above method, the calculation of the advantage function in multiplayer games can also be implemented with pre-trained predictor similar to the two-player setting.

## 5. Experiment Results and Discussion

We evaluate the performance of A-PSRO in multiple game environments. We select the state-of-the-art game solvers as baselines, including PSRO (Lanctot et al., 2017), Pipeline-PSRO (P-PSRO) (McAleer et al., 2020), DPP-PSRO (Perez-Nieves et al., 2021), UDF-PSRO (Liu et al., 2021) and PSD-PSRO (Yao et al., 2023). To ensure a fair comparison, all other components of the PSRO framework are kept unchanged, with strategy exploration being the only aspect that differs among the algorithms.

### 5.1. Experiments in Symmetric Zero-sum Games

In symmetric zero-sum games, we test A-PSRO with both LookAhead and diversity modules on complex real-world games. The environment we chose for testing is the normal-form games generalized used in the previous PSRO algorithms (Czarnecki et al., 2020; Liu et al., 2022).

In Figure 2, we show the results in typical zero-sum games. Additional experiments are presented in the Supplementary Material. From Figure 2, we can see that our method achieves a notable reduction in exploitability across all game environments, sometimes by several orders of magnitude. A-PSRO without diversity module outperforms A-PSRO in the Transitive game. The reason is that the diversity module is designed to navigate the constraints of non-transitive structure, and its impact is limited in games with strong tran-

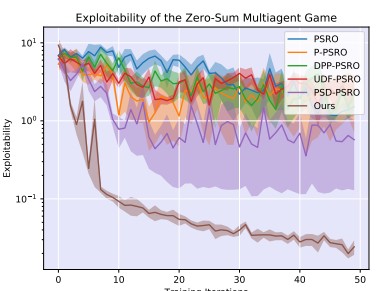
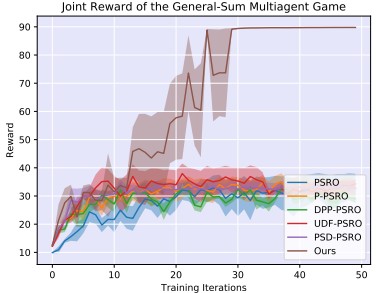

(a) Exploitability of Zero-sum Games  (b) Joint Reward of General-sum Games

*Figure 6.* The exploitability of zero-sum multiagent games and joint reward of general-sum multiagent games. Each algorithm is tested in 4 randomly generated identically distributed game environments, and the averages are plotted.

sitive dimensions. The effectiveness of A-PSRO without the LookAhead module has a significant decline in all the games, which indicates that our LookAhead module greatly contributes to approximating Nash equilibria in all games.

Compared to diversity-based algorithms, A-PSRO exhibits higher exploitability during the early stages, primarily due to subgames failing to fully cover the entire strategy space. This observation underscores the importance of incorporating diversity exploration into the learning process. When combined with diversity exploration, A-PSRO achieves a stable and rapid reduction in exploitability, even in scenarios where other algorithms experience stagnation.

In Figure 3, we show the distribution of advantages in different games. We use the non-linear dimensionality reduction method t-SNE (t-Distributed Stochastic Neighbor Embedding) (Van der Maaten & Hinton, 2008) to map the strategies into the unit matrix and maintain adjacency between strategies. In games with a strong transitive dimension (AlphaStar, Transitive game), the advantage function exhibits rapid changes around the Nash equilibrium. Conversely, in games with a strong cyclic dimension, when the advantage function changes slowly, the diversity module becomes crucial for the learning process of the Nash equilibrium.

We also compare the running time of A-PSRO with other PSRO algorithms. The experimental results and settings are given in the Supplementary Material for detailed analysis.

### 5.2. Experiments in Two-player General-sum Games

It is worth noting that the compared algorithms do not guarantee convergence in all general-sum games. To ensure a fair comparison, we extended the zero-sum game structure to a general-sum game with multiple equilibria and verified that all algorithms successfully converged to equilibrium under this setting. The detailed game structure is given in the Supplementary Material.

In Figures 4(a) and 4(b), we present the training results

of the algorithms in the aforementioned games. Figure 4 demonstrates that A-PSRO consistently learns the optimal equilibrium strategy, whereas other algorithms acquire different equilibria and often stagnate in suboptimal equilibria.

We further conduct experiments in randomly generated games, and the results are depicted in Figure 4(c). In our experiments, A-PSRO also attains the highest reward.

We depict the distribution of advantage in the aforementioned games in Figure 5. It is evident from the figure that the advantage of general-sum games is non-convex. Algorithms based on the strategy gradient are likely to converge to different equilibrium strategies from various initial points.

In general-sum games, A-PSRO requires exploring multiple equilibrium oracles to identify higher-reward strategies. Compared to other algorithms, this inevitably leads to increased computational complexity. We will address this limitation in the future work.

### 5.3. Experiments in Multi-player Games

For multi-player games, we test our method in both zero-sum and general-sum games. We randomly generated a series of identically distributed games as test environments.

Figure 6(a) illustrates the distance of the learned strategies from the Nash equilibrium for different algorithms in multi-player zero-sum games. As shown in the figure, A-PSRO effectively explores strategies with higher advantage and converges toward the Nash equilibrium, whereas other algorithms exhibit slower convergence.

Figure 6(b) presents the joint rewards of the algorithms during the training process of multi-player general-sum games. The results align with Figure 4(c), indicating that A-PSRO consistently learns the equilibrium strategy with the highest joint reward. The parameter setting and experimental details are given in the Supplementary Material.

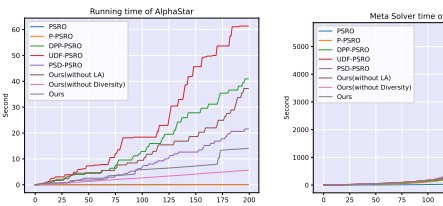

(a) Strategy exploration time    (b) Meta game solver time

*Figure 7.* The running time of different algorithms. The left figure shows the cumulative time spent by different algorithms during a single strategy exploration given a meta-equilibrium. The right figure shows the cumulative time spent by different algorithms in solving the equilibrium of the meta-game.

### 5.4. Comparison of Computational Complexity

In this section, we compare the computational complexity of A-PSRO and diversity-based PSRO algorithms.

Assume that the payoff $U$ is a $[n, n]$ matrix, and population $P_i$ and $P_j$ are $[p, n]$ matrixs. The current meta-equilibrium $\pi$ is an $[n, 1]$ vector, and the update step size is $d$.

Taking the classic EC diversity metric as an example:

$$\mathrm{EC}\left(\mathcal{P}_i \mid \mathcal{P}_j\right) := \mathrm{Tr}\left(\mathbf{I} - (\mathcal{L} + \mathbf{I})^{-1}\right)$$
$$\mathcal{L} = \mathcal{M}_i \mathcal{M}_i^T, \mathcal{M}_i = \mathcal{P}_i \times U_i \times \mathcal{P}_j$$

Its computational complexity is $O(pn^2 + p^2n + p^3)$. The strategy exploration process requires the exploration of each update directions in pure strategy space to get the one that maximizes diversity. Thus the actual computational complexity is $n \times O(pn^2 + p^2n + p^3) = O(pn^3 + p^2n^2 + p^3n)$.

Regarding the computation of the advantage function, the following presents an implementation method we use in our code. First, repeat $\pi$ and derive a $[n, n]$ matrix $Q$, and then the LookAhead update direction can be obtained through:

$$\min([Q \cdot (1 - d) + I \cdot d] \times U \times I).\mathrm{argmax}().$$

This process has a computational complexity of $O(n^3)$, which is independent of the population size, and lower than the complexity of diversity-based exploration.

In our experiments, the time-consuming modules include meta-game solving, diversity-based strategy exploration, and non-diversity-based strategy exploration. Among them, the experimental code only differs in the last module between A-PSRO and other algorithms.

From Figure 7(a), we can see that if only the LookAhead module is used (ours without diversity), the time spent on strategy exploration in A-PSRO increases almost linearly. From other algorithms (which perform diversity exploration with a certain probability), it can be observed that diversity

exploration leads to a nonlinear increase in the time per iteration. This suggests that using the advantage function as an evaluation metric does not introduce more computational complexity compared to diversity metrics.

From Figure 7(b) and empirical analysis, it can be observed that the solving time of the meta-game with fictitious play is an exponential function of the population size. A-PSRO has the longest runtime, indicating that A-PSRO has the largest population size during training. Considering that in the pipeline improvement (McAleer et al., 2020), the PSRO algorithm does not expand the population at every iteration but only adds new strategies when the existing ones converge (see Algorithm 2 in appendix for details), this demonstrates that A-PSRO's strategy exploration quickly improves the existing strategies in the population to optimal.

**Comparison with Traditional Game Solver.**   The above results also demonstrate the advantages of population-based equilibrium solving algorithms over traditional approaches. Compared to exact solution algorithms such as linear programming, PSRO-type algorithms can obtain approximate solutions at lower computational cost. If we consider an exploitability level of $10^{-2}$ to be sufficiently close to equilibrium, A-PSRO requires fewer than 50 iterations to achieve this, with a total runtime less than 1 minute, which is significantly lower than the time required for exact solutions.

On the populations obtained from A-PSRO, fictitious Play only requires about $10^3$ of iterations to reach exploitability $10^{-4}$. In contrast, it takes about $10^4$ iterations directly using fictitious play. Since the process of solving meta-equilibria based on fictitious play is the most time-consuming component in the PSRO framework, A-PSRO accelerates the convergence speed of the PSRO framework toward Nash equilibria through efficient strategy exploration.

### 6. Conclusion

In this paper, we introduce A-PSRO, a unified open-ended framework for learning equlibrium strategies. We propose the advantage function as an evaluation metric for the strategy. The advantage function exhibits favorable properties, such as convexity and Lipschitz continuity. Leveraging the advantage function, A-PSRO effectively enhances the objective of strategy exploration during population expansion. In zero-sum games, A-PSRO can deterministically approach Nash equilibrium strategies during iterations, significantly reducing the exploitability of learned strategies. Moreover, in general-sum games with multiple equilibria, A-PSRO maximizes rewards during the learning of Nash equilibria. Experimental results demonstrate the robust generalization capabilities of A-PSRO as an open-ended framework in large-scale environments, highlighting its potential to advance equilibrium theory in multiagent systems.

## Acknowledgements

This paper is supported by National Key R&D (Research and Development) Programe of China (2021YFA1000403), the National Natural Science Foundation of China (No.12431012).

## Impact Statement

This paper presents work whose goal is to advance the field of PSRO framework for solving equlibrium strategy. There are many potential societal consequences of our work, none which we feel must be specifically highlighted here.

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

# A. Omitted Proofs

## A.1. Proof of Theorem 3.1

*Theorem.* In symmetric zero-sum games, if the joint strategy $(\pi_i^1, \pi_j^2)$ is a Nash equilibrium, we have $(\pi_i^1, \pi_j^1)$ and $(\pi_i^2, \pi_j^2)$ are both Nash equilibriums.

*Proof.* From the definition, the joint strategy $(\pi_i^1, \pi_j^2)$ is a Nash equilibrium implies that the exploitability $\mathcal{E}(\pi_i^1, \pi_j^2) = 0$. Then we will have:

$$\mathcal{E}(\pi_i^1, \pi_j^2) = \max_{\pi_i'}[U_i(\pi_i', \pi_j^2) - U_i(\pi_i^1, \pi_j^2)] + \max_{\pi_j'}[U_j(\pi_i^1, \pi_j') - U_j(\pi_i^1, \pi_j^2)] = 0. \tag{1}$$

This indicates that:

$$\max_{\pi_i'}[U_i(\pi_i', \pi_j^2) - U_i(\pi_i^1, \pi_j^2)] = \max_{\pi_j'}[U_j(\pi_i^1, \pi_j') - U_j(\pi_i^1, \pi_j^2)] = 0. \tag{2}$$

Then we prove that $U_i(\pi_i^1, \pi_j^2) = U_j(\pi_i^1, \pi_j^2) = 0$. If the reward of agents are not all 0, since the game is zero-sum, we assume that:

$$U_i(\pi_i^1, \pi_j^2) > U_j(\pi_i^1, \pi_j^2). \tag{3}$$

Since the game is symmetric, we will have:

$$U_i(\pi_i^1, \pi_j^1) = U_j(\pi_i^1, \pi_j^1) = 0. \tag{4}$$

This indicates that:

$$\max_{\pi_j'}[U_j(\pi_i^1, \pi_j') - U_j(\pi_i^1, \pi_j^2)] \neq 0, \tag{5}$$

which leads to contradiction. Therefore, we prove that the rewards of both agents are 0.

From the equations above we have that $U_j(\pi_i^1, \pi_j^1) = U_j(\pi_i^1, \pi_j^2) = 0$. Since $\pi_j^2$ is a best response to $\pi_i^1$, we can see that $\pi_j^1$ is also a best response to $\pi_i^1$. This indicates that $(\pi_i^1, \pi_j^1)$ and $(\pi_i^2, \pi_j^2)$ are both Nash equilibriums. $\square$

## A.2. Proof of Theorem 3.2

*Theorem.* For any two-player game, when the strategy of another player is fixed (denoted as $\pi_j$), there always exists pure strategy $a_i \in \mathcal{A}$ which satisfies that $a_i \in \mathrm{BR}(\pi_j)$. Particularly, in zero-sum games, $U_i(\pi_i, \pi_j)$ is always the same for all $\pi_j \in \mathrm{BR}(\pi_i)$.

*Proof.* We assume that there exists strategy $\pi_i^* = (p_1, \cdots, p_{|\mathcal{A}|})$ which is the best response of $\pi_j$. Then we have:

$$U_i(\pi_i^*, \pi_j) = p_1 U_i(a_1, \pi_j) + \cdots + p_{|\mathcal{A}|} U_i(a_{|\mathcal{A}|}, \pi_j) \leq \max_{l \in \{1, \cdots, |\mathcal{A}|\}} U_i(a_l, \pi_j), \tag{6}$$

which implies that $a_l \in \mathcal{A}$ is a best response to $\pi_j$.

In zero-sum games, we assume that the strategy of the player $i$ is fixed as $\pi_i$. Then we have:

$$\mathrm{BR}(\pi_i) = \mathrm{argmax}_{\pi_j} U_j(\pi_i, \pi_j). \tag{7}$$

We assume that for all $\pi_j \in \mathrm{BR}(\pi_i)$, the reward of player $j$ is $U_j(\pi_i, \pi_j) = U^0$. If the game is zero-sum, the reward of player $i$ is $U_i(\pi_i, \pi_j) = -U^0$. This implies that in zero-sum games, $U_i(\pi_i, \pi_j)$ is always the same for all $\pi_j \in \mathrm{BR}(\pi_i)$. $\square$

## A.3. Proof of Theorem 3.4

*Theorem.* In two-player zero-sum games,

- $\mathcal{E}(\pi_i, \pi_j) = -(\mathcal{V}_i(\pi_i) + \mathcal{V}_j(\pi_j))$.

- $\mathcal{V}_i(\pi_i)$ is Lipschitz continuous about $\pi_i$, and $-\mathcal{V}_i(\pi_i)$ is a convex function about $\pi_i$.

- If the game is symmetric, $\forall \pi_i$, $\mathcal{V}_i(\pi_i) \leq 0$. The joint strategy $(\pi_i^0, \pi_j^0)$ is a Nash equilibrium if and only if $\mathcal{V}_i(\pi_i^0) = \mathcal{V}_j(\pi_j^0) = 0$. In games with only transitive dimension, $\mathcal{V}_i(\pi_i) > \mathcal{V}_j(\pi_j)$ implies $U_i(\pi_i, \pi_j) > 0$.

*Proof.*
- From the definition, we can easily find that $\mathcal{E}(\pi_i, \pi_j) = -(\mathcal{V}_i(\pi_i) + \mathcal{V}_j(\pi_j))$. We define the domain of strategies $D = \{(p_1, p_2, \cdots, p_{|\mathcal{A}|})\}$ which satisfies that $p_k \geq 0$, $\sum_{k=1}^{|\mathcal{A}|} p_k = 1$, and $|\mathcal{A}|$ is the dimension of the action space $\mathcal{A} = \{a_1, \cdots, a_{|\mathcal{A}|}\}$.

- In order to prove that $-\mathcal{V}_i(\pi_i)$ is convex function about $\pi_i$, we need to prove that for $\pi_i^1, \pi_i^2 \in D$, and $c \in (0, 1)$, we have $-\mathcal{V}_i[(1-c)\pi_i^1 + c\pi_i^2] \leq -(1-c)\mathcal{V}_i(\pi_i^1) - c\mathcal{V}_i(\pi_i^2)$.

We assume that $\pi_i^1 = (p_1^1, \cdots, p_{|\mathcal{A}|}^1)$ and $\pi_i^2 = (p_1^2, \cdots, p_{|\mathcal{A}|}^2)$. Then we have:

$$
\begin{aligned}
\mathcal{V}_i[(1-c)\pi_i^1 + c\pi_i^2] &= U_i[(1-c)\pi_i^1 + c\pi_i^2, a_0], \quad a_0 \in \mathcal{A} \cap \text{BR}[(1-c)\pi_i^1 + c\pi_i^2] \\
&= (1-c)U_i(\pi_i^1, a_0) + cU_i(\pi_i^2, a_0) \\
&\geq (1-c)U_i(\pi_i^1, a_0^1) + cU_i(\pi_i^2, a_0^2), \quad a_0^t \in \mathcal{A} \cap \text{BR}(\pi_i^t), \ t \in \{1, 2\} \\
&= (1-c)\mathcal{V}_i(\pi_i^1) + c\mathcal{V}_i(\pi_i^2).
\end{aligned}
\tag{8}
$$

This implies that the inverse function of advantage function $-\mathcal{V}_i(\pi_i)$ is convex about $\pi_i$.

Then we prove that $\mathcal{V}_i(\pi_i)$ is Lipschitz continuous about $\pi_i$. We assume that $\pi_i = (p_1, \cdots, p_{|\mathcal{A}|})$, $\pi_i' = \pi_i + \Delta\pi_i = (p_1 + \Delta p_1, \cdots, p_{|\mathcal{A}|} + \Delta p_{|\mathcal{A}|})$ which satisfies that $\sum_{k=1}^{|\mathcal{A}|} \Delta p_k = 0$, and $a_0' \in \mathcal{A} \cap \text{BR}(\pi_i + \Delta\pi_i)$, $a_0 \in \mathcal{A} \cap \text{BR}(\pi_i)$.

$$
\begin{aligned}
&\mathcal{V}_i(p_1 + \Delta p_1, \cdots, p_{|\mathcal{A}|} + \Delta p_{|\mathcal{A}|}) - \mathcal{V}_i(p_1, \cdots, p_{|\mathcal{A}|}) \\
=&U_i(\pi_i + \Delta\pi_i, a_0') - U_i(\pi_i, a_0) \\
=&(p_1 + \Delta p_1)U_i(a_1, a_0') + \cdots + (p_{|\mathcal{A}|} + \Delta p_{|\mathcal{A}|})U_i(a_{|\mathcal{A}|}, a_0') - p_1 U_i(a_1, a_0) - \cdots - p_{|\mathcal{A}|}U_i(a_n, a_0) \\
=&A_1 \Delta p_1 + \cdots + A_{|\mathcal{A}|}\Delta p_{|\mathcal{A}|} + [U_i(\pi_i, a_0') - U_i(\pi_i, a_0)]
\end{aligned}
\tag{9}
$$

where $A_1, \cdots, A_m$ are constants.

Then we will prove that there exists $M$ such that $\forall \delta > 0$, the following conclusion holds:

$$
|U_i(\pi_i, a_0') - U_i(\pi_i, a_0)| < \delta M, \quad \text{when } \max|\Delta p_k| \leq \delta.
\tag{10}
$$

First we consider that $a_0' \in \mathcal{A} \cap \text{BR}(\pi_i)$, this indicates that $|U_i(\pi_i, a_0') - U_i(\pi_i, a_0)| = 0$.

Then we consider that $a_0' \notin \mathcal{A} \cap \text{BR}(\pi_i)$, which means that:

$$
U_i(\pi_i, a_0) > U_i(\pi_i, a_0')
\tag{11}
$$

We prove that $U_i(\pi_i, a)$ is Lipschitz continuous with respect to $\pi_i$. $\forall \pi_i, \forall a_m$, we have:

$$
\begin{aligned}
U_i(\pi_i + \Delta\pi_i, a_m) - U_i(\pi_i, a_m) &= \sum_{k=1}^{|\mathcal{A}|}(p_k + \Delta p_k)U_i(a_k, a_m) - \sum_{k=1}^{|\mathcal{A}|} p_k U_i(a_k, a_m) \\
&= \Delta p_1 U_i(a_1, a_m) + \cdots + \Delta p_{|\mathcal{A}|}U_i(a_{|\mathcal{A}|}, a_m) \\
&\leq \delta M \quad \text{where } \delta = \max|\Delta p_k| \text{ and } M = \max_{a^* \in \mathcal{A}} |\sum_{i=1}^{|\mathcal{A}|} U_i(a_i, a^*)|.
\end{aligned}
\tag{12}
$$

Since we are considering games with finite dimension, M has consistent upper bounds. Then we have:

$$
U_i(\pi_i, a_0) + \delta M \geq U_i(\pi_i', a_0) \geq U_i(\pi_i', a_0') \geq U_i(\pi_i, a_0') - \delta M.
\tag{13}
$$

Using Equation (12) twice results in (13). This indicates that $0 \leq U_i(\pi_i, a_0') - U_i(\pi_i, a_0) \leq 2\delta M$. According to Equation (9), we have

$$
\mathcal{V}_i(p_1 + \Delta p_1, \cdots, p_{|\mathcal{A}|} + \Delta p_{|\mathcal{A}|}) - \mathcal{V}_i(p_1, \cdots, p_{|\mathcal{A}|}) \leq \max_k A_k \cdot \delta \cdot |\mathcal{A}| + 2\delta M
\tag{14}
$$

This means that $\mathcal{V}_i(\pi_i)$ is Lipschitz continuous about $\pi_i$.

- From Theorem 1 we can easily find that if the game is symmetric, $\mathcal{V}_i(\pi_i) \leq 0$. The joint strategy $(\pi_i^1, \pi_i^1)$ is a Nash equilibrium if and only if $\mathcal{V}_i(\pi_i^1) = 0$.

In games with only transitive dimension, the best response is the same $\pi_j^0$ for all strategies $\pi_i$. Then we have:

$$\mathcal{E}(\pi_i^1, \pi_j^0) = -\mathcal{V}_i(\pi_i^1) < -\mathcal{V}_i(\pi_i^2) = \mathcal{E}(\pi_i^2, \pi_j^0) \tag{15}$$

So $\pi^1$ is closer to the Nash equilibrium than $\pi^2$ in the transitive dimension, which means that $U_i(\pi_i^1, \pi_j^2) > 0$.

$\square$

### A.4. Proof of Theorem 4.1

*Theorem.* In symmetric zero-sum games, given the population $\mathcal{P}_i = \mathcal{P}_j = \{\pi_i^1, \cdots, \pi_i^t\}, \forall \pi_i^k \in \mathcal{P}_i$, we have $\mathcal{V}_i(\pi_i^k) \leq \mathcal{V}_i(\theta_i \mid \mathcal{P}_i)$. Here, $\theta_i$ is the equilibrium of the meta-game corresponding to the population $\mathcal{P}_i$.

*Proof.* The population $\mathcal{P}_i$ can be viewed as a subgame. Applying Theorem 3.4 for this subgame we get the following property:

$$\forall \pi_i^k \in \mathcal{P}_i, \ \mathcal{V}_i(\pi_i^k \mid \mathcal{P}_i) \leq \mathcal{V}_i(\theta_i \mid \mathcal{P}_i). \tag{16}$$

Then we the following derivation:

$$\begin{aligned}
\mathcal{V}_i(\pi_i^k) &= U_i(\pi_i^k, \mathrm{BR}(\pi_i^k)) = -U_i(\mathrm{BR}(\pi_i^k), \pi_i^k) \\
&\leq -U_i(\mathrm{BR}^*(\pi_i^k), \pi_i^k) = U_i(\pi_i^k, \mathrm{BR}^*(\pi_i^k)) \quad \text{where } \mathrm{BR}^*(\pi_i^k) = \mathrm{argmin}_{\pi' \in \mathcal{P}_i} U_i(\pi_i^k, \pi'). \\
&\leq U_i(\theta_i, \mathrm{BR}^*(\theta_i)) \\
&= \mathcal{V}_i(\theta_i \mid \mathcal{P}_i).
\end{aligned} \tag{17}$$

From this theorem, we can see that $\mathcal{V}_i(\theta_i \mid \mathcal{P}_i)$ is an upper bound for $\mathcal{V}_i(\pi_i^k)$. This suggests that exploring new strategies in the neighborhood of $\theta_i$ increase the probability of improving the advantage of population. $\square$

Here we give an example of $\mathcal{V}_i(\pi_i^k) \geq \mathcal{V}_i(\theta_i)$.

|       | $a_1$      | $a_2$      | $a_3$      | $a_4$       | $a_5$       | $a_6$       |
|-------|------------|------------|------------|-------------|-------------|-------------|
| $a_1$ | (0,0)      | (1,-1)     | (-1,1)     | (-0.1,0.1)  | (0.9,-0.9)  | (-1.1,1.1)  |
| $a_2$ | (-1,1)     | (0,0)      | (1,-1)     | (-1.1,1.1)  | (-0.1,0.1)  | (0.9,-0.9)  |
| $a_3$ | (1,-1)     | (-1,1)     | (0,0)      | (0.9,-0.9)  | (-1.1,1.1)  | (-0.1,0.1)  |
| $a_4$ | (0.1,-0.1) | (1.1,-1.1) | (-0.9,0.9) | (0,0)       | (1,-1)      | (-1,1)      |
| $a_5$ | (-0.9,0.9) | (0.1,-0.1) | (1.1,-1.1) | (-1,1)      | (0,0)       | (1,-1)      |
| $a_6$ | (0.9,-0.9) | (1.1,-1.1) | (0.1,-0.1) | (1,-1)      | (-1,1)      | (0,0)       |

In this game, assuming that the current population is $\mathcal{P}_i = \mathcal{P}_j = \{a_1, a_5\}$, then $\mathcal{V}_i(a_5) = -1$. However, $\mathcal{V}_i(\theta_i \mid \mathcal{P}_i) = \mathcal{V}_i(a_1) = -1.1 < -1$.

### A.5. Proof of Theorem 4.2

*Theorem.* Given the meta-equilibrium strategy $\theta_i$, if $\mathcal{V}_i(\theta_i) < 0$, there exists $\Delta \pi_i \in \mathcal{A}$ and $\delta > 0$ satisfying:

$$\forall \, 0 < d < \delta, \quad \mathcal{V}_i\left((1-d) \cdot \theta_i + d \cdot \Delta \pi_i\right) > \mathcal{V}_i(\theta_i).$$

*Proof.* From Theorem 3.4 we have that the inverse of advantage function $-\mathcal{V}_i(\pi_i)$ is convex about $\pi_i$. Since $\mathcal{V}_i(\theta_i) < 0 = \max \mathcal{V}_i(\pi_i)$, from the convexity of the function $-\mathcal{V}_i$ we can find a direction of descent in the domain of the strategy $\mathcal{D} = \{(p_1, \cdots, p_{|\mathcal{A}|}) \mid p_m \geq 0, \sum p_m = 1\}$:

$$\exists \, \delta', \pi' \in \mathcal{D}, \, \forall \, 0 < d < \delta', \quad \mathcal{V}_i((1-d) \cdot \theta_i + d \cdot \pi') > \mathcal{V}_i(\theta_i). \tag{18}$$

Since the domain of the strategy $\mathcal{D}$ is a convex combination of pure strategy space $\mathcal{A}$, $\pi' \in \mathcal{D}$ can also be expressed as a convex combination of all elements in $\mathcal{A}$. This means that there exists $\Delta \pi_i \in \mathcal{A}$, which satisfies that $\langle \pi', \Delta \pi_i \rangle > 0$. We define that $\delta = \frac{|\langle \pi', \Delta \pi_i \rangle|}{|\Delta \pi_i| \cdot |\pi'|} \cdot \delta'$. Then we have:

$$\forall\, 0 < d < \delta, \quad \mathcal{V}_i((1-d) \cdot \theta_i + d \cdot \Delta \pi_i) > \mathcal{V}_i(\theta_i). \tag{19}$$

$\square$

### A.6. Proof of Theorem 4.4

*Theorem.* In two-player simplified games, $\forall \pi_i$, for any $a_j^l \in \operatorname{argmax}_{a_j \in \mathrm{BR}(\pi_i) \cap \mathcal{A}_j} U_i(\pi_i, a_j)$ and $\forall \delta > 0$, there always exists $\pi_i'$ which satisfies $|\pi_i' - \pi_i| < \delta$ and $\mathrm{BR}(\pi_i') \cap \mathcal{A}_j = \{a_j^l\}$.

*Proof.* We assume that $\pi_i = (p_1, \cdots, p_{|\mathcal{A}_i|})$. Since we are searching for $\pi_i'$ in the neighbour of $\pi_i$, without loss of generality, we assume $p_t > 0$, $\forall t \in \{1, \cdots, |\mathcal{A}_i|\}$. Since the game is simplified, the pure strategy $a_j^l$ is not dominated. Therefore, there exists $m \in \{1, \cdots, |\mathcal{A}_i|\}$, $\forall a_j' \neq a_j^l$, $a_j' \cdot U_j \cdot a_i^m < a_j^l \cdot U_j \cdot a_i^m$. We choose $\pi_i' = (\frac{1-(1+\delta) \cdot p_m}{1-p_m} p_1, \cdots, (1+\delta)p_m\cdot, \cdots, \frac{1-(1+\delta) \cdot p_m}{1-p_m} p_{|\mathcal{A}_i|})$, it is obvious that $|\pi_i' - \pi_i| < \delta$. Then we proof $\forall a_j' \neq a_j^l$, $a_j' \in \mathrm{BR}(\pi_i)$, we have $U_j(\pi_i', a_j^l) > U_j(\pi_i', a_j')$. It is obvious that

$$U_j(\pi_i, a_j^l) - U_j(\pi_i, a_j') = p_1(a_j^l \cdot U_j \cdot a_i^1 - a_j' \cdot U_j \cdot a_i^1) + \cdots + p_{|\mathcal{A}_i|}(a_j^l \cdot U_j \cdot a_i^{|\mathcal{A}_i|} - a_j' \cdot U_j \cdot a_i^{|\mathcal{A}_i|}) = 0 \tag{20}$$

Then we have

$$U_j(\pi_i', a_j^l) - U_j(\pi_i', a_j') = \frac{1-(1+\delta) \cdot p_m}{1-p_m} p_1(a_j^l \cdot U_j \cdot a_i^1 - a_j' \cdot U_j \cdot a_i^1)$$

$$+ \cdots + (1+\delta)p_m(a_j^l \cdot U_j \cdot a_i^m - a_j' \cdot U_j \cdot a_i^m) + \cdots + \frac{1-(1+\delta) \cdot p_m}{1-p_m} p_{|\mathcal{A}_i|}(a_j^l \cdot U_j \cdot a_i^{|\mathcal{A}_i|} - a_j' \cdot U_j \cdot a_i^{|\mathcal{A}_i|}) \tag{21}$$

$$= \delta \cdot \frac{p_m}{1-p_m}(a_j^l \cdot U_j \cdot a_i^m - a_j' \cdot U_j \cdot a_i^m) > 0$$

This indicates that $\mathrm{BR}(\pi_i') \cap \mathcal{A}_j = \{a_j^l\}$. $\square$

### A.7. Proof of Theorem 4.6

*Theorem.* In two-player simplified games,

- $\forall i$, $\mathcal{V}_i(\pi_i)$ is Lipschitz continuous.

- We assume that the joint strategy $(\pi_i, \pi_j)$ is a Nash equilibrium. If $\mathrm{BR}(\pi_i) \cap \mathcal{A}_j$ has the unique element, then $\mathcal{V}_i(\pi_i)$ is a local maximum.

- Under the same assumption, if $(\pi_i^1, \pi_j^2)$ and $(\pi_i^3, \pi_j^4)$ are both NEs, then $(\pi_i^1, \pi_j^2)$ Pareto dominates $(\pi_i^3, \pi_j^4)$ if and only if $\mathcal{V}_i(\pi_i^1) \geq \mathcal{V}_i(\pi_i^3)$ and $\mathcal{V}_j(\pi_j^2) \geq \mathcal{V}_j(\pi_j^4)$.

*Proof.*
- In Theorem 3.4, the proof of the Lipschitz continuity of the advantage function does not require that the game is zero-sum. Therefore, we can similarly prove that $\mathcal{V}_i(\pi_i)$ is Lipschitz continuous in two-player general-sum games.

- If the joint strategy $(\pi_i, \pi_j)$ is a Nash equilibrium, we have that $\pi_j \in \mathrm{BR}(\pi_i)$. We assume that:

$$\mathrm{BR}(\pi_i) \cap \mathcal{A}_j = \{a_j^0\}, \tag{22}$$

which means that:

$$\forall a_j^k \neq a_j^0,\ U_j(\pi_i, a_j^k) < U_j(\pi_i, a_j^0). \tag{23}$$

From the proof of Theorem 3.4 we have that $\forall \pi_i, \forall a$, $U_i(\pi_i, a)$ is Lipschitz continuous about $\pi_i$. Then there must exsits $\delta > 0$, which satisfies that:

$$\forall \pi_i' \in B_\delta(\pi_i),\ (\mathrm{BR}(\pi_i') \cap \mathcal{A}_j) \subseteq (\mathrm{BR}(\pi_i) \cap \mathcal{A}_j), \tag{24}$$

where $B_\delta(\pi_i)$ is the open ball of radius $\delta$ centered on $\pi_i$. From Theorem 3.2, we have that $\{a_j^0\} = \mathrm{BR}(\pi_i') \cap \mathcal{A}_j$, then we have:

$$
\begin{aligned}
\mathcal{V}_i(\pi_i) = U_i(\pi_i, \pi_j) &= U_i(\pi_i, a_j^0) \\
&\geq U_i(\pi_i', \pi_j) = U_i(\pi_i', a_j^0) \quad (\text{becaue}(\pi_i, \pi_j)\text{is Nash equilibrium}) \\
&= \mathcal{V}_i(\pi_i').
\end{aligned}
\tag{25}
$$

We assume that the elements of $\mathrm{BR}(\pi_i) \cap \mathcal{A}_j$ are unique in assumption. Since $(\pi_i, \pi_j)$is Nash equilibria, $\pi_j$ is the optimal response to $\pi_i$. This indicates that $\pi_j = a_j^0$ ($\pi_j$ is linear combination of $\mathrm{BR}(\pi_i) \cap \mathcal{A}_j$). Therefore the third holds. This illustrates that $\mathcal{V}_i(\pi_i)$ is a local maximum of the advantage function $\mathcal{V}_i$.

- Under the assumption that $\mathrm{BR}(\pi_i) \cap \mathcal{A}_j$ are unique, we have $\mathcal{V}_i(\pi_i^1) = U_i(\pi_i^1, \pi_j^2)$ and $\mathcal{V}_j(\pi_j^2) = U_j(\pi_i^1, \pi_j^2)$. Therefore, $(\pi_i^1, \pi_j^2)$ Pareto dominate $(\pi_i^3, \pi_j^4)$ is equivalent to $U_i(\pi_i^1, \pi_j^2) \geq U_i(\pi_i^3, \pi_j^4)$ and $U_j(\pi_i^1, \pi_j^2) \geq U_j(\pi_i^3, \pi_j^4)$. Since $(\pi_i^1, \pi_j^2)$ and $(\pi_i^3, \pi_j^4)$ are both Nash equilibrium, this holds if and only if $\mathcal{V}_i(\pi_i^1) \geq \mathcal{V}_i(\pi_i^3)$ and $\mathcal{V}_j(\pi_j^2) \geq \mathcal{V}_j(\pi_j^4)$.

$\square$

## A.8. Proof of Theorem 4.7

*Theorem.* In two-player simplified games, the current population for agent $i$ is $\mathcal{P}_i = \{\pi_i^1, \cdots, \pi_i^t\}$. $\theta_i$ is the global maximum point of the advantege $\mathcal{V}_i$ in $\mathrm{hull}(\mathcal{P}_i)$. Then there must exist a non-zero measure set $\mathcal{D}' \subset \mathrm{hull}(\mathcal{P}_i)$, which satisfies that if $\theta_i'$ is a local maximum point of the advantege $\mathcal{V}_i$ in $\mathcal{D}'$, then $\mathcal{V}_i(\theta_i') = \mathcal{V}_i(\theta_i)$.

*Proof.* We assume that the strategy of the player $i$ is $\pi_i = (p_1, \cdots, p_{|\mathcal{A}_i|})$. For $k \in \{1, \cdots, |\mathcal{A}_j|\}$, we define $g_k(\pi_i) = U_j(\pi_i, a_j^k)$, where $a_j^k \in \mathcal{A}_j$. Then we have:

$$
g_k(p_1, \cdots, p_{|\mathcal{A}_i|}) = p_1 U_j(a_i^1, a_j^k) + \cdots + p_{|\mathcal{A}_i|} U_j(a_i^{|\mathcal{A}_i|}, a_j^k).
\tag{26}
$$

If the elements in set $\mathrm{BR}(\pi_i) \cap \mathcal{A}_j$ are not unique, there must exists $k$ and $k'$ which satisfies that $g_k(\pi_i) = g_{k'}(\pi_i)$. Since the game is simplified, $\left( U_j(a_i^1, a_j^k), \cdots, U_j(a_i^{|\mathcal{A}_i|}, a_j^k) \right)$ and $\left( U_j(a_i^1, a_j^{k'}), \cdots, U_j(a_i^{|\mathcal{A}_i|}, a_j^{k'}) \right)$ are linearly independent vectors. This indicates that $\pi_i$ satisfying $g_k(\pi_i) = g_{k'}(\pi_i)$ is a zero measure set and non-dense in the domain $\Pi_i$.

We define

$$
D^0 = \{\pi_i \mid \pi_i \in \mathrm{hull}(\mathcal{P}_i), (\mathrm{BR}(\pi_i) \cap \mathcal{A}_j) \text{ is a singleton set}\}.
\tag{27}
$$

Since $\pi_i$ satisfying that there exists $k$ and $k'$ with $g_k(\pi_i) = g_{k'}(\pi_i)$ is non-dense in the domain $\Pi_i$, we consider the projection of those $\pi_i$ onto $\mathrm{hull}(\mathcal{P}_i)$. This indicates that either $D^0$ is an empty set (which means that $\pi_i$ intersects different functions $g$ covers the plane of $\mathrm{hull}(\mathcal{P}_i)$), or $(\mathrm{hull}(\mathcal{P}_i) \setminus D^0)$ is a non-dense set.

$D^0$ is an empty set means that $\forall \pi_i \in \mathrm{hull}(\mathcal{P}_i)$, $g_k(\pi_i) = g_{k'}(\pi_i)$. Since the game is simplified, $\forall \pi_i \in \mathrm{hull}(\mathcal{P}_i)$, $U_i(\pi_i, a_j^k) = U_i(\pi_i, a_j^{k'})$. Thus, we can remove $a_j^{k'}$ feom $\mathcal{A}_j$, which does not affect the calculation of $\mathcal{V}_i(\pi_i)$. This is because in this theorem, $g_k$ and $g_{k'}$ are always the same on $\mathrm{hull}(\mathcal{P})$ when $D^0$ is an empty set. Since all operations in this theorem are performed on $\mathrm{hull}(\mathcal{P})$, and both $g$ correspond to the same $U_j$, by the definition of $\mathcal{V}$, it is only necessary to keep the one corresponding to the larger $U_i$. Therefore, we can remove another one for simplification.

If $(\mathrm{hull}(\mathcal{P}_i) \setminus D^0)$ is a non-dense set, we consider separately whether $\theta_i \in D^0$. If $\theta_i \in D^0$, there must exists non-zero measure set $\mathcal{D}^1 \subseteq \mathrm{hull}(\mathcal{P}_i)$, which satisfies that:

$$
\forall \pi_i \in \mathcal{D}^1, \ \mathrm{BR}(\pi_i) \cap \mathcal{A}_j = \mathrm{BR}(\theta_i) \cap \mathcal{A}_j.
\tag{28}
$$

This indicates that $\mathcal{V}_i(\pi_i)$ is a linear function about $\pi_i$. Since $\theta_i$ is the global maximum of this linear function, there must exists non-zero measure set $\mathcal{D}' \subseteq \mathrm{hull}(\mathcal{P}_i)$, which satisfies that if $\theta_i'$ is a local maximum in $\mathcal{D}'$, then $\mathcal{V}_i(\theta_i') = \mathcal{V}_i(\theta_i)$.

If $\theta_i \in (\mathrm{hull}(\mathcal{P}_i) \setminus D^0)$, we assume that:

$$
\mathrm{argmax}_k \, g_k(\theta_i) = \{1, \cdots, l\}.
\tag{29}
$$

Due to the Lipschitz continuity of $U_j$, there must exists $d > 0$, which satisfies that:

$$
\forall \Delta \pi_i, \ \mathrm{BR}(\theta_i^* + \frac{\Delta \pi_i}{|\Delta \pi_i|} \cdot d) \subseteq \{\pi_j^1, \cdots, \pi_j^l\}.
\tag{30}
$$

Since $g_k(\pi_i)$ is a hyperplane corresponds to a linear function, the value of function $g_k(\pi_i)$ on the ray $\theta_i + \frac{\Delta \pi_i}{|\Delta \pi_i|} \cdot \delta$, $0 < \delta < d$ is either all maximal or all non-maximal for all $k$. Thus we can assume that there is a unique pure strategy best response $\pi_j^k, k \in \{1, \cdots, l\}$ for a strategy on that ray.

Thus, the open ball $B_d(\theta_i)$ can be divided into at most $|\mathcal{A}_j|$ region $D^k, k \in \{1, \cdots, |\mathcal{A}_j|\}$. In every region $D^k$, $\mathcal{V}_i(\pi_i)$ is linear function about $\pi_i$. Since $\theta_i$ is the global maximum of this linear function, there must exists non-zero measure set $\mathcal{D}' \subseteq B_d(\theta_i)$, which satisfies that if $\theta_i'$ is a local maximum in $\mathcal{D}'$, then $\mathcal{V}_i(\theta_i') = \mathcal{V}_i(\theta_i)$. $\qquad \square$

### A.9. Proof of Theorem 4.8

*Theorem.* Assuming that $|\nabla_{\pi_i} \hat{\mathcal{V}}_i(\pi_i) - \nabla_{\pi_i} \mathcal{V}_i(\pi_i)| \leq \frac{1}{3} |\nabla_{\pi_i} \mathcal{V}_i(\pi_i)|$, the algorithm will converge to equilibrium with sublinear convergence rate in symmetric zero-sum games.

*Proof.* We use $x$ to denote $\pi_i$ and $f(x)$ to denote $\mathcal{V}_i(\pi_i)$. We use $f^*$ to denote the global maximum of the function $f$. According to Theorem 4, $f(x)$ is Lipschitz continuous about $x$, and $-f(x)$ is a convex function about $x$. From the convexity, there exists $M$ for $\forall \eta$,

$$
\begin{aligned}
f(x - \eta \nabla \hat{f}(x)) &= f(x) - \eta \nabla \hat{f}(x)^T \nabla f(x) + \frac{M}{2} |-\eta \nabla \hat{f}(x)|^2 \\
&\leq f(x) - \frac{2\eta}{3} |\nabla f(x)|^2 + \frac{M}{2} |-\eta \cdot \frac{4}{3} \nabla f(x)|^2 \\
&= f(x) - (\frac{2\eta}{3} - \frac{8M\eta^2}{9}) |\nabla f(x)|^2.
\end{aligned}
\tag{31}
$$

By assuming that $\eta \leq \frac{9}{48M}$, we have

$$
\frac{2\eta}{3} - \frac{8M\eta^2}{9} \geq \frac{\eta}{2}
\tag{32}
$$

Then we have:

$$
f(x - \eta \nabla \hat{f}(x)) \leq f(x) - \frac{\eta}{2} |\nabla f(x)|^2
\tag{33}
$$

We denote $\bar{x} = x - \eta \nabla \hat{f}(x)$, then we have:

$$
\begin{aligned}
f(\bar{x}) &\leq f(x) - \frac{\eta}{2} |\nabla f(x)|^2 \\
&\leq f^* + \nabla f(x)^T (x - x^*) - \frac{\eta}{2} |\nabla f(x)|^2 \\
&= f^* + \frac{1}{2\eta} \left( \|x - x^*\|^2 - \|x - x^* - \eta \nabla f(x)\|^2 \right) \\
&= f^* + \frac{1}{2\eta} \left( \|x - x^*\|^2 - \|\tilde{x} - x^*\|^2 \right)
\end{aligned}
\tag{34}
$$

$$
\begin{aligned}
\sum_{i=1}^{k} \left( f\left(x^i\right) - f^* \right) &\leq \frac{1}{2\eta} \sum_{i=1}^{k} \left( \|x^{i-1} - x^*\|^2 - \|x^i - x^*\|^2 \right) \\
&= \frac{1}{2\eta} \left( \|x^0 - x^*\|^2 - \|x^k - x^*\|^2 \right) \\
&\leq \frac{1}{2\eta} \|x^0 - x^*\|^2.
\end{aligned}
\tag{35}
$$

It is obvious that $f(x^i)$ is non-increasing, then we have:

$$
f\left(x^k\right) - f^* \leq \frac{1}{k} \sum_{i=1}^{k} \left( f\left(x^i\right) - f^* \right) \leq \frac{1}{2k\eta} \|x^0 - x^*\|^2
\tag{36}
$$

The convergence rate is $\mathcal{O}(\frac{1}{k})$. This indicates that the approximate gradient-based algorithm will converge to the equilibrium with sublinear convergence.

$\square$

# B. Algorithm Introduction and the Pseudo-Code

## B.1. Fictitious Play

In fictitious play algorithm with two players, strategies are randomly initialized as $(\pi_i^0, \pi_j^0)$. In iteration $t$, agents select the best response to the average strategy of its opponent:

$$\pi_i^{t+1} = \mathrm{BR}(\bar{\pi}_j^t), \quad \bar{\pi}_j^t = \frac{1}{t} \sum_{k=1}^{t} \pi_j^k. \tag{37}$$

Fictitious play has convergence guarantees in simple structures such as two-player zero-sum games. However, it has the disadvantage that convergence can be very slow in games with large strategy spaces.

## B.2. Classic PSRO Algorithm

---
**Algorithm 1** Policy-Space Response Oracles
---
**Input**: initial policy populations for all players $\mathcal{P}$. Compute the expected utilities $U^{\mathcal{P}}$ for each joint $\pi \in \mathcal{P}$. Initialize meta-strategies $\theta_i = \mathrm{UNIFORM}(\mathcal{P}_i)$

  1: **while** iters $e$ in $\{1, 2, \cdots\}$ **do**
  2:    **for** player $i \in \{1, \cdots, n\}$ **do**
  3:      **for** many episodes **do**
  4:        Sample $\pi_{-i} \sim \theta_{-i}$
  5:        Train oracle $\pi_i'$ over $\mathcal{O}(\pi_i', \pi_{-i})$
  6:      **end for**
  7:      $\mathcal{P}_i = \mathcal{P}_i \cup \{\pi_i'\}$
  8:    **end for**
  9:    Compute missing entries in $U^{\mathcal{P}}$ from $\mathcal{P}$
10:    Compute a meta-strategy $\theta$ from $U^{\mathcal{P}}$
11: **end while**
**Output**: Current solution strategy $\theta_i$ for player $i$.

---

Pseudo-code of the classic PSRO algorithm is given in Algorithm 1. UNIFORM denotes random sampling according to the uniform distribution. The two main components of the algorithm are the exploration of the new strategy $\pi_i'$ and the computation of the meta-strategy $\theta$. In this paper, we focus on improving the PSRO framework from the perspective of new strategy exploration. We use the meta-strategy solver with exactly the same parameters as the other PSRO algorithms in our comparison experiments (Perez-Nieves et al., 2021; Liu et al., 2021).

## B.3. A-PSRO for Solving Zero-Sum Games

In this section, we provide the algorithm for the generation of new strategy, and the rest of the framework is the same as other PSRO algorithms. We assume that the current population is $(\mathcal{P}_i, \mathcal{P}_j)$, where $\mathcal{P}_i = \{\pi_i^1, \cdots, \pi_i^t\}$. Since A-PSRO primarily improves the strategy exploration process, we outline how to enhance strategies through exploration within a single PSRO iteration. The other components of the algorithm remain consistent with the PSRO algorithms compared in this study.

It is worth noting that the PSRO variant algorithms typically prioritize updating existing strategies. New strategies are generated randomly only when the existing ones fail to improve. For further details, please refer to the DPP-PSRO or PSD-PSRO algorithms. Similarly, in our algorithm, the agents initially decide to update the last strategy $\pi_i^t$. The new strategy $\pi_i^{t+1}$ is generated and incorporated into the population only if the update process does not enhance the utility. As LookAhead updates the strategy in the transitive dimension, we set its learning rate lower than $\|\theta_i\|^{\infty}$ to prevent stagnation as the strategy approaches the Nash equilibrium.

---

**Algorithm 2** Strategy exploration process of A-PSRO in zero-sum games

---

**Input**: Population $(\mathcal{P}_i, \mathcal{P}_j)$, meta-Nash equilibrium $(\theta_i, \theta_j)$, strategy to be updated of the agent $\pi_i^t$.
**Parameter**: diversity weight $\lambda_d$, learning rate $l_r$, improvement bound $c_m$.

 1: Randomly generate $d_r \sim \mathbf{U}[0, 1]$.
 2: **if** $d_r \le \lambda_d$ **then**
 3:    $\Delta\pi = \mathrm{argmax}_{\Delta\pi \in \mathcal{A}} \left[ \mathrm{EC}(\mathcal{P}_i \setminus \{\pi_i^t\} \cup \{(1 - l_r) \cdot \pi_i^t + l_r \cdot \Delta\pi\} \mid \mathcal{P}_j) \right]$.
 4:    $\pi_i^* = (1 - l_r) \cdot \pi_i^t + l_r \cdot \Delta\pi$.
 5: **else**
 6:    Randomly generate $l_r \sim \mathbf{U}[0, \min(l_r, \|\theta_i\|^\infty)]$.
 7:    $\Delta\pi = \mathrm{argmax}_{\Delta\pi \in \mathcal{A}} \mathcal{V}_i[(1 - l_r) \cdot \theta_i + l_r \cdot \Delta\pi]$.
 8:    $\pi_i^* = (1 - l_r) \cdot \theta_i + l_r \cdot \Delta\pi$.
 9: **end if**
10: **if** $\frac{\pi_i^* \times \mathcal{U}_i \times \theta_j}{\pi_i^t \times \mathcal{U}_i \times \theta_j} - 1 \ge c_m$ **then**
11:    $\pi_i^t = \pi_i^*$.
12: **else**
13:    $\pi_i^t = \pi_i^*$. Then randomly generate $\pi_i^{t+1}$, $\mathcal{P}_i = \mathcal{P}_i \cup \{\pi_i^{t+1}\}$. (The randomly generated strategy $\pi_i^{t+1}$ will be updated in the next iteration, this is equivalent to adding an explored strategy.)
14: **end if**
15: **return** $\mathcal{P}_i$.

**Output**: $\mathcal{P}_i$

---

In Algorithm 2, we combine the diversity module and our LookAhead module. The EC (expected cardinality) function is the diversity measure used in (Perez-Nieves et al., 2021):

$$
\begin{aligned}
\mathrm{EC}(\mathcal{P}_i \mid \mathcal{P}_j) &:= \mathrm{Tr}(\mathbf{I} - (\mathcal{L} + \mathbf{I})^{-1}) \\
\mathcal{L} &= \mathcal{M}_i \mathcal{M}_i^T, \ \mathcal{M}_i = \mathcal{P}_i \times U_i \times \mathcal{P}_j.
\end{aligned}
\tag{38}
$$

Here Tr denotes the trace of a matrix. In all zero-sum game experiments, we control the proportion of diversity and LookAhead modules with a uniform parameter $\lambda_d$. We find from our experimental results that A-PSRO achieves good convergences in games with different transitive and cyclic structures.

### B.4. A-PSRO for Solving Two-Player General-Sum Games

In experiments with general-sum games, we find that the diversity module does not contribute to improving the reward of the strategy learning process. Therefore, the strategy exploration process of our algorithm A-PSRO contains only the LookAhead module. The Pseudo-Code of A-PSRO is given in Algorithm 3. In Algorithm 3, the meta-solver of the oracle $\mathcal{O}(\mathcal{P}_i, \mathcal{P}_j \mid \pi_{i,j}^k)$ is the fictitous play with 1000 itereations. In our experiments, we set the number of repeats $k = 10$. The rest of the A-PSRO algorithm in the general-sum game is consistent with the zero-sum game.

### B.5. A-PSRO for Solving Multi-Player Games

The main modification in applying A-PSRO algorithm to solve multi-player games is the computation of the advantage function. Unlike the two-player game with direct utilization of the best response BR, the computation of the advantege of the strategy $\pi_i$ requires the usage of oracle $\mathcal{O}(\Pi_{-i} \mid \pi_i)$. In multi-player games, we adopt the joint best response as an approximation to the optimistic equilibrium. The Pseudo-Code of calculating the advantage in A-PSRO is given in Algorithm 4. Besides the computation of the advantage function, A-PSRO is consistent with the two-player game in the multi-player game.

---

**Algorithm 3** Strategy exploration process of A-PSRO in general-sum games

---

**Input**: Population $(\mathcal{P}_i, \mathcal{P}_j)$, meta-Nash equilibrium $(\theta_i, \theta_j)$, strategy to be updated of the agent $\pi_i^t$.
**Parameter**: learning rate $l_r$, improvement bound $c_m$.

1: **for** repeats $k$ in $\{1, 2, \cdots\}$ **do**
2:      $(\pi_i^k, \pi_j^k) = \text{UNIFORM}[\text{hull}(\mathcal{P}_{i,j})]$
3:      $(\theta_i^k, \theta_j^k) = \mathcal{O}(\mathcal{P}_i, \mathcal{P}_j \mid \pi_{i,j}^k)$
4: **end for**
5: $\theta_i = \text{argmax}_k \, \mathcal{V}_i(\theta_i^k)$
6: Randomly generate $l_r \sim \mathbf{U}[0, \min(l_r, \|\theta_i\|^\infty)]$.
7: $\Delta\pi = \text{argmax}_{\Delta\pi \in \mathcal{A}} \, \mathcal{V}_i[(1 - l_r) \cdot \theta_i + l_r \cdot \Delta\pi]$.
8: $\pi_i^* = (1 - l_r) \cdot \theta_i + l_r \cdot \Delta\pi$.
9: **if** $\frac{\pi_i^* \times \mathcal{U}_i \times \theta_j}{\pi_i^t \times \mathcal{U}_i \times \theta_j} - 1 \geq c_m$ **then**
10:      $\pi_i^t = \pi_i^*$.
11: **else**
12:      $\pi_i^t = \pi_i^*$. Then randomly generate $\pi_i^{t+1}$, $\mathcal{P}_i = \mathcal{P}_i \cup \{\pi_i^{t+1}\}$.
13: **end if**
14: **return** $\mathcal{P}_i$.

**Output**: $\mathcal{P}_i$

---

**Algorithm 4** Calculation of the advantage function in multi-player games

---

**Input**: Strategy of the player $\pi_i$, initializaion $U_i^0 = -M, U_{-i}^0 = -M$.

1: The identifying numbers set of other agents is $\{-i\} = \{1, \cdots, k\}$
2: The joint pure strategy space of other agents is $\mathcal{A}_{-i} = \mathcal{A}_1 \times \cdots \times \mathcal{A}_k$
3: **for** $(a_1^m, \cdots, a_k^m) \in \mathcal{A}_{-i}$ **do**
4:      **if** $U_{-i}(\pi_i, \pi_{-i} = (a_1^m, \cdots, a_k^m)) > U_{-i}^0$ **then**
5:          $U_i^0 = U_i(\pi_i, \pi_{-i}), U_{-i}^0 = U_{-i}(\pi_i, \pi_{-i})$
6:      **else if** $U_{-i}(\pi_i, \pi_{-i} = (a_1^m, \cdots, a_k^m)) = U_{-i}^0$ **then**
7:          **if** $U_i(\pi_i, \pi_{-i} = (a_1^m, \cdots, a_k^m)) > U_i^0$ **then**
8:              $U_i^0 = U_i(\pi_i, \pi_{-i})$
9:          **end if**
10:      **end if**
11: **end for**
12: $\mathcal{V}_i(\pi_i) = U_i^0$

**Output**: $\mathcal{V}_i(\pi_i)$

---

| Settings | Value | Description |
|---|---|---|
| nb_iters | 200 | Training iterations |
| meta_solver | fictitious play | Metasolver method |
| meta_iter | 1000 | Iterations for Metasolver |
| improvement_threshold $c_m$ | 0.03 | Convergence criteria |
| learning_rate | 0.5 | Default learning rate |
| num_learners | 4 | Number of strategies updated in each iteration |
| num_repeats | 10 | Number of repetitions per experiment |
| $l_r$ | 0.5 | Default step size |
| $\lambda_d$ | 0.5 | Diversity weight |

*Table 1.* Parameter setting for experiments in Zero-sum games.

## C. Experiment Details and Additional Experiment Results

### C.1. A-PSRO for Solving Zero-Sum Games

The parameter setting of zero-sum games is given in Table 1. All experiments in this paper were run with CPU support on model Intel Core i9-10900KF CPU @ 3.70GHz. Experiments can be performed under both Windows and Linux systems.

In our setup, each experiment is repeated 4 times and the results are averaged for plotting. Within each experiment, the population is initialized randomly, and the meta-game is solved in the same manner for different algorithms. The number of learners for all algorithms, except the classic PSRO, is set to 4, which implies that four strategies within the population will be updated in each iteration. In order to more accurately compare the effiency of different algorithms in learning the strategies, we gradually increase the iterations for meta-solver. The initial iterations for meta-solver is set as 1000. Every 20 steps of training, we increase the iterations for meta-solver by 500.

Our experiments for symmetric zero-sum games are conducted in the environments used in the previous papers about PSRO algorithms. Detailed description of these game environments can be found in (Czarnecki et al., 2020; Liu et al., 2022). Taking the AlphaStar as an example, it is derived from the experimental environment StarCraft, which is commonly used in multiagent reinforcement learning. By extracting meta-strategies in large scale extend-form game StarCraft, we can obtain a symmetric normal-form game AlphaStar. In detail, AlphaStar is a symmetric zero-sum games with dimension $888 \times 888$. Other normal-form game environments are similarly obtained by extracting meta-strategies for real world games (Go, Kuhn Poker, etc.).

Additional experiment results are shown in Figure 8. According to the previous work (Czarnecki et al., 2020), the following games (8(c),8(e),8(f),8(g),8(h),8(n)) has strong transitive structures. From Figure 8, we can see that in these games, adopting only the LA (lookahead) module with the objective of maximizing the advantage function is effective to reduce the exploitability. In those games with cyclic structures, it is necessary to adopt the diversity module in learning Nash equilibrium. We can see that A-PSRO combining LA and Diversity Module achieves the optimal results across all environments. In the stochastic game Disc game 8(d) with almost no transitive dimension, all algorithms obtain the same convergence results.

Figure 9 shows the advantage distribution of these games. From Figure 9, we can see that although there may be large differences in the payoff matrices between different games, they may have similar advantage distribution. Games with the same advantage distribution have similar convergence processes of strategies when applying the PSRO algorithms to solve them, e.g. 9(c), 9(g), 9(n). Although the advantage function does not fully characterize the transitive dimension in zero-sum games, we believe that it has similarities to the geometric visualization of the transitive and cyclic dimensions in the previous work (Czarnecki et al., 2020).

### C.2. A-PSRO for Solving Two-Player General-Sum Games

The parameter setting of general-sum games is given in Table 2. The hardware and system setup used for the experiments are the same as those for zero-sum games. Similar to zero-sum games, the initial iterations for meta-solver is set as 1000. Every 20 steps of training, we increase the iterations for meta-solver by 500.

The StagHunt game is a commomly used general-sum game environment to test the ability of algorithms to learn the optimal Nash equilibrium (Tang et al., 2021). The payoff matrix of the traditional StagHunt game is given in Table 3. In the Stag

| Settings | Value | Description |
|---|---|---|
| nb_iters | 100 | Training iterations |
| meta_solver | fictitious play | Metasolver method |
| meta_iter | 1000 | Iterations for Metasolver |
| num_oracle_repeats $k$ | 10 | Repetitions of the inner loop for strategy exploration |
| distribution_type | normal | Gaussian distribution |
| distribution_mean | 0 | Mean value of the distribution |
| distribution_var | 20 | Variance of the distribution |
| improvement_threshold | 0.03 | Convergence criteria |
| learning_rate | 0.5 | Default learning rate |
| num_learners | 4 | Number of strategies updated in each iteration |
| num_repeats | 100 | Number of repetitions per experiment |

*Table 2.* Parameter setting for experiments in General-sum games.

Hunt game, both (U,L) and (D,R) are Nash equilibriums. In order to achieve a higher reward joint strategy, cooperation is required in the learning process of agents.

|  | L | R |
|---|---|---|
| U | 30,30 | -10,-10 |
| D | -10,-10 | 20,20 |

*Table 3.* Traditional StagHunt game.

|  | $a_1$ | $a_2$ | $\cdots$ | $a_i$ | $\cdots$ | $a_n$ |
|---|---|---|---|---|---|---|
| $a_1$ | $U_1$ | -u | -u | -u | -u | -u |
| $a_2$ | -u | u | $\cdots$ | -u | $\cdots$ | u |
| $\vdots$ | -u | $\cdots$ | $\ddots$ | -u | $\ddots$ | $\vdots$ |
| $a_i$ | -u | -u | -u | $U_i$ | -u | -u |
| $\vdots$ | -u | $\vdots$ | $\ddots$ | -u | $\ddots$ | $\vdots$ |
| $a_n$ | -u | u | $\cdots$ | -u | $\cdots$ | u |

*Table 4.* Advanced StagHunt game.

In this paper, we extend the traditional StagHunt game structure to large scale general-sum game. The payoff matrix of Advanced-Staghunt is given in Table C.2. In Advanced-Staghunt, each agent has a pure strategy space $\{a_1 \cdots , a_n\}$, where $\{a_1, a_i, \cdots\}$ corresponds to the Nash equilibrium strategies resulting from cooperation. In Table C.2, $U_i$ denotes the reward corresponding to the cooperative strategy in StagHunt, which is drawn from a unifrom ditribution $\mathbf{U}[1, 2]$. In order to judge whether A-PSRO deterministically converges to the optimal Nash equilibrium, we set one of those $U_i$ to 2.

For the rest of the payoff matrix for the Advanced-StagHunt, we use the uniform distribution $u = \mathbf{U}[0, 0.8]$ to fill the rewards corresponding to each joint strategy. This suggests that there are many inefficient Nash equilibria in the Advanced-StagHunt game besides the cooperative equilibrium in the shape of $(a_i, a_i)$.

In our experiments, we set $n = 100$ and there are 5 cooperation equilibrium with reward $U_i \sim \mathbf{U}[1, 2]$. Each PSRO algorithm was run 10 times repeatedly to solve the game and the results were averaged for presentation.

From the experiment result 4(a), we can see that most PSRO algorithms in the Advanced-StagHunt stagnate in the inefficient Nash equilibria. This is because the space of strategies whose strategy gradient points to a cooperative Nash equilibrium is a small proportion of the full space. In order for the agent to learn the optimal Nash equilibrium strategy, it is necessary to design reward-related objective for the agent. We can see that A-PSRO based on the advantage function effectively learns the optimal Nash equilibrium strategy, which indicates that the strategy exploration objective with advantage can improve rewards when solving general-sum games.

The optimal Nash equilibrium in the Advanced-StagHunt game is a pure strategy equilibrium. In order to test the effectiveness

of A-PSRO in games where the optimal equilibrium is a mixed strategy equilibrium, we design a large-scale general-sum game Advanced-RSP with the structure similar to the traditional game Rock-Paper-Scissors. We first design the structure of the RSP in general-sum game $U_{RSP}$. The payoff matrix of $U_{RSP}$ is given in Table 5. In $U_{RSP}$, $\epsilon$ is a random number satisfying the uniform distribution $\mathbf{U}[0, 100]$. This general-sum game has the similar mixed Nash equilibrium to the traditional RSP.

Based on the $U_{RSP}$, we design the large scale general-sum game Advanced-RSP. The payoff matrix of Advanced-RSP is given in Table C.2. For the rest of the payoff matrix for the Advanced-RSP, we use the uniform distribution $u = \mathbf{U}[0, 100]$ to fill the rewards corresponding to each joint strategy. We can easily find that each subgame corresponding to the joint strategy $(R_i, S_i, P_i)$ is a mixed Nash equilibrium. There are also other equilibra with inefficient rewards.

In our experiment, we set $n = 1000$ and $i = 10$. Each PSRO algorithm was run 10 times repeatedly to solve the game and the results were averaged for presentation. The experiment result is shown in Figure 4(b). From the figure, we can see that A-PSRO learns the optimal mixed equilibrium strategy, while all other PSRO algorithms stagnate in the inefficient Nash equilibrium.

|  | R | S | P |
|---|---|---|---|
| R | 100 | 180+$\epsilon$ | $\epsilon$ |
| S | $\epsilon$ | 100 | 180+$\epsilon$ |
| P | 180+$\epsilon$ | $\epsilon$ | 100 |

Table 5. General-sum RSP structure $U_{RSP}$.

|  | $a_1$ | $\cdots$ | $R_1$ | $S_1$ | $P_1$ | $\cdots$ | $a_j$ | $\cdots$ | $R_i$ | $S_i$ | $P_i$ | $\cdots$ | $a_n$ |
|---|---|---|---|---|---|---|---|---|---|---|---|---|---|
| $a_1$ | $u$ | $\cdots$ | -$u$ | -$u$ | -$u$ | $\cdots$ | $u$ | $\cdots$ | -$u$ | -$u$ | -$u$ | $\cdots$ | $u$ |
| $\vdots$ | $\vdots$ | $\ddots$ | -$u$ | -$u$ | -$u$ | $\ddots$ | $\vdots$ | $\ddots$ | -$u$ | -$u$ | -$u$ | $\ddots$ | $\vdots$ |
| $R_1$ | -$u$ | -$u$ | | | | -$u$ | -$u$ | -$u$ | -$u$ | -$u$ | -$u$ | -$u$ | -$u$ |
| $S_1$ | -$u$ | -$u$ | | $U_{RSP}$ | | -$u$ | -$u$ | -$u$ | -$u$ | -$u$ | -$u$ | -$u$ | -$u$ |
| $P_1$ | -$u$ | -$u$ | | | | -$u$ | -$u$ | -$u$ | -$u$ | -$u$ | -$u$ | -$u$ | -$u$ |
| $\vdots$ | $\vdots$ | $\ddots$ | -$u$ | -$u$ | -$u$ | $\ddots$ | $\vdots$ | $\ddots$ | -$u$ | -$u$ | -$u$ | $\ddots$ | $\vdots$ |
| $a_j$ | $u$ | $\cdots$ | -$u$ | -$u$ | -$u$ | $\cdots$ | $u$ | $\cdots$ | -$u$ | -$u$ | -$u$ | $\cdots$ | $u$ |
| $\vdots$ | $\vdots$ | $\ddots$ | -$u$ | -$u$ | -$u$ | $\ddots$ | $\vdots$ | $\ddots$ | -$u$ | -$u$ | -$u$ | $\ddots$ | $\vdots$ |
| $R_i$ | -$u$ | -$u$ | -$u$ | -$u$ | -$u$ | -$u$ | -$u$ | -$u$ | | | | -$u$ | -$u$ |
| $S_i$ | -$u$ | -$u$ | -$u$ | -$u$ | -$u$ | -$u$ | -$u$ | -$u$ | | $U_{RSP}$ | | -$u$ | -$u$ |
| $P_i$ | -$u$ | -$u$ | -$u$ | -$u$ | -$u$ | -$u$ | -$u$ | -$u$ | | | | -$u$ | -$u$ |
| $\vdots$ | $\vdots$ | $\ddots$ | -$u$ | -$u$ | -$u$ | $\ddots$ | $\vdots$ | $\ddots$ | -$u$ | -$u$ | -$u$ | $\ddots$ | $\vdots$ |
| $a_n$ | $u$ | $\cdots$ | -$u$ | -$u$ | -$u$ | $\cdots$ | $u$ | $\cdots$ | -$u$ | -$u$ | -$u$ | $\cdots$ | $u$ |

Table 6. Advanced RSP game.

We also perform experiments in randomly generated games that feature the same reward distribution. The dimension of random generated games are $1000 \times 1000$. In these games, each element of the payment matrix is generated by a normal distribution with mean $\mu = 0$ and variance $\sigma^2 = 20$. To test the convergence results of different algorithms in random generated games, we let the PSRO algorithms operate in 100 independently generated game environments and the results were averaged for presentation. The experiment result is shown in Figure 4(c). From Figure 4(c), we can see that A-PSRO achieves the optimal reward of the joint strategy.

Since A-PSRO requires multiple equilibrium oracles in general-sum games to explore strategies with higher rewards, its runtime increases significantly compared to existing algorithms. In future work, we will explore ways to simplify this process.

| Settings | Value | Description |
|---|---|---|
| players | 3 | Number of agents in the game |
| nb_iters | 50 | Training iterations |
| meta_solver | fictitious play | Metasolver method |
| meta_iter | 10000 | Iterations for Metasolver |
| distribution_type | normal | Gaussian distribution |
| distribution_mean | 0 | Mean value of the distribution |
| distribution_var | 20 | Variance of the distribution |
| improvement_threshold | 0.03 | Convergence criteria |
| learning_rate | 0.5 | Default learning rate |
| num_learners | 4 | Number of strategies updated in each iteration |
| num_repeats | 4 | Number of repetitions per experiment |
| $\lambda_d$ | 0.5 | Diversity weight |

*Table 7.* Parameter setting for experiments in multi-player games.

### C.3. A-PSRO for Solving Multi-Player Games

The parameter setting of multi-player games is given in Table 7. The hardware and system setup used for the experiments are the same as those for zero-sum games. In our multi-player game experiments, we adopt randomly generated games that feature the same reward distribution.

In multi-player zero-sum games, we use randomly generated symmetric games with dimension $20 \times 20 \times 20$. This is because we have found in our experiments that reducing exploitability in larger-scale games requires very large computational complexity. We believe that a comparison with other algorithms in the setting of this size is sufficient to demonstrate effectiveness. During the generation of these games, we added constraints to avoid generating strong pure strategies, which substantially increased the difficulty of strategy learning.

In the generation of multi-player general-sum games, we use the structure similar to the Advanced-Staghunt game with dimension $10 \times 10 \times 10$, and set the reward of the optimal equilibrium strategy to 90.

## D. Code and Dataset

We provide part of the code necessary for the full operation of A-PSRO in the supplementary materials. Once the paper is accepted, we will upload the complete A-PSRO code along with the game data used for testing.

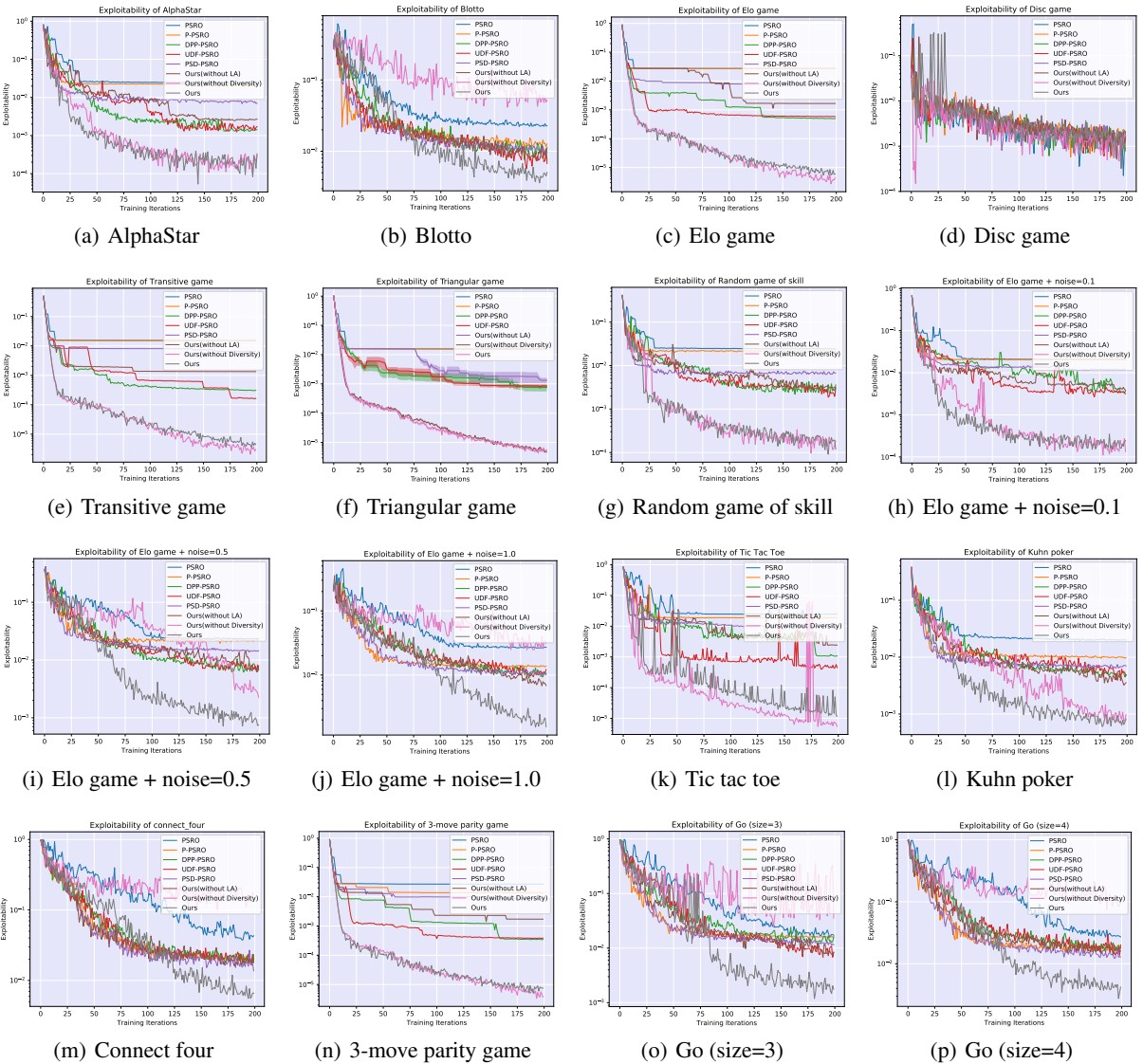

*Figure 8.* The exploitability of the joint strategy learned by agents in various zero-sum games is depicted. The reduction in exploitability through population iterations can serve as an indicator of the effectiveness in approximating the Nash equilibrium.

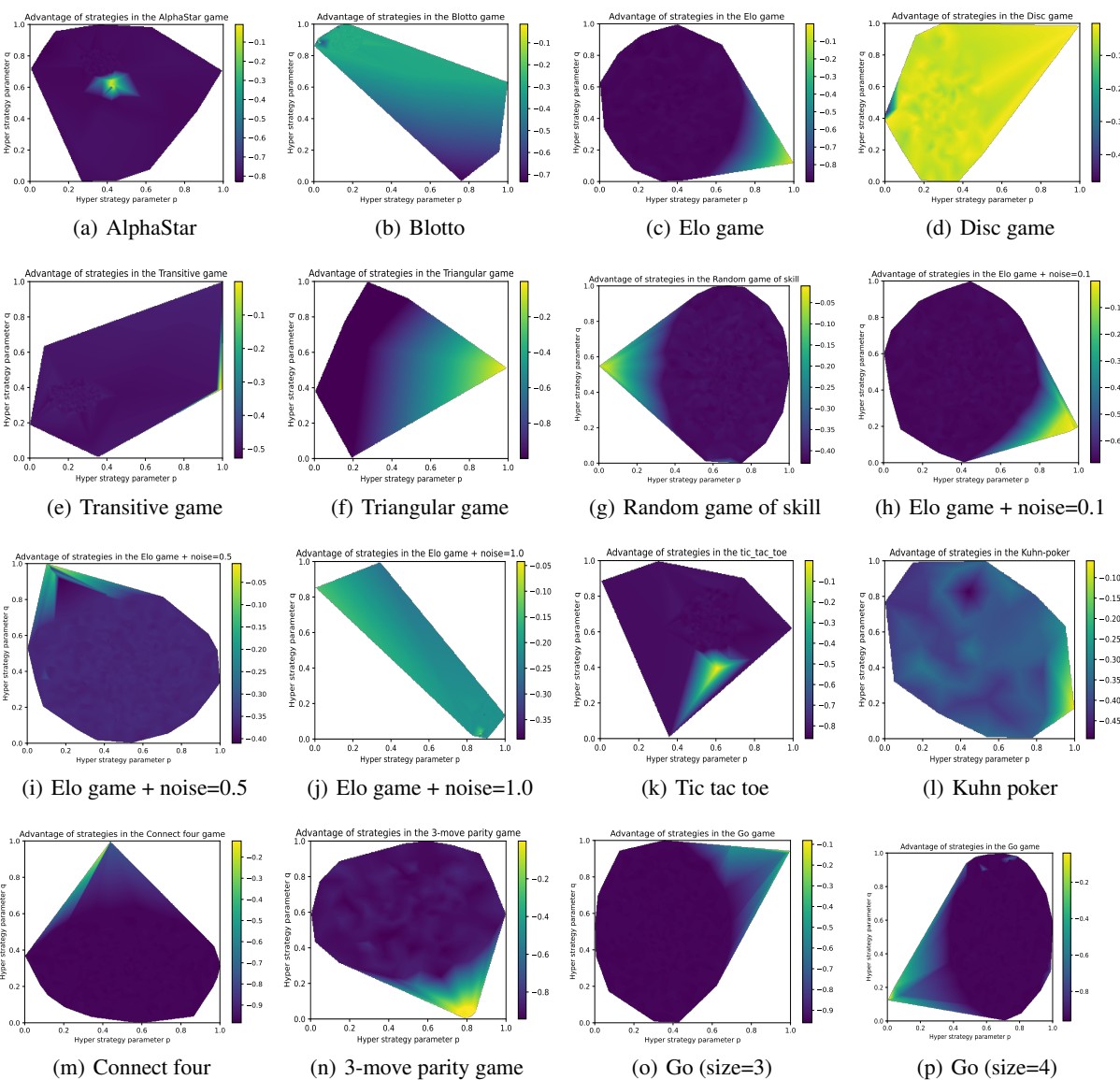

*Figure 9.* The advantage distribution of strategies in various zero-sum games. Lighter colored regions indicate strategies with higher advantage.

