# OpenReview forum: "A-PSRO: A Unified Strategy Learning Method with Advantage Metric for Normal-form Games"
_ICML.cc/2025/Conference — ICML 2025 poster_

### Official Review · Reviewer_ayi4 · 2025-03-11

**Overall Recommendation:** 3

**Summary:**

This paper proposes Advantage Policy Space Response Oracle (A-PSRO), a new framework for learning Nash equilibria in normal-form games with large strategy spaces, applicable to both zero-sum and general-sum settings. The key contribution is the Advantage function, a new evaluative metric that guides strategy updates toward equilibrium, ensuring efficient and deterministic convergence in zero-sum games while optimizing for higher-reward equilibria in general-sum games. To enhance learning efficiency, A-PSRO introduces LookAhead updates for faster equilibrium approximation and meta-equilibrium search to identify high-reward strategies. Finally, empirical results demonstrate that A-PSRO reduces exploitability more effectively in zero-sum settings and achieves superior rewards in general-sum games, offering a scalable and unified solution for strategic learning.

**Claims And Evidence:**

All claims are well-supported by both theoretical analysis and experiments, particularly for zero-sum games.

**Essential References Not Discussed:**

No, I did not find any essential related works that are missing from the paper.

**Experimental Designs Or Analyses:**

Yes, I reviewed the experimental design and analysis. The evaluation includes zero-sum and general-sum normal-form games, using exploitability as the primary metric for assessing convergence to Nash equilibrium. The experiments compare A-PSRO against state-of-the-art PSRO variants (e.g., P-PSRO, DPP-PSRO, UDF-PSRO, PSD-PSRO) across various game environments, including AlphaStar, Go, Staghunt, and Randomly Generated Games. The experimental setup is generally sound, as it uses well-established benchmarks and relevant baselines.

**Methods And Evaluation Criteria:**

Yes, the proposed method generally make sense for the problem of learning Nash equilibria in norm-form games.

**Other Comments Or Suggestions:**

In Page 4, Theorem 3.4 states that $-V_i(\pi_i)$ is a convex function. However, above Theorem 4.1, the paper states that “Since $V_i(\pi_i) $ is convex……”, which may be a typo.

**Other Strengths And Weaknesses:**

Strengths:

1. The introduction of the Advantage function as a new strategy evaluation metric is a novel idea that extends beyond traditional best-response and diversity-based approaches in PSRO.

2. The LookAhead update mechanism offers a deterministic strategy improvement method, which differentiates it from existing stochastic diversity-based strategy exploration methods

3. The paper is well-structured, with clear theoretical explanations and empirical validation.

Weakness:

1. The paper does not explicitly discuss the computational overhead of Advantage function evaluations compared to standard PSRO methods. A runtime comparison or complexity analysis would strengthen claims about scalability.

**Questions For Authors:**

1. How does the additional computation required for Advantage function evaluation and LookAhead updates compare to standard PSRO methods? Could you provide a complexity analysis or runtime comparison?

2. I know this paper focuses on normal-form games. Since many real-world applications (e.g., poker, security games) involve extensive-form or imperfect-information settings, I just wonder whether A-PSRO can be extended to handle sequential decision-making scenarios? If so, what modifications would be needed?

**Relation To Broader Scientific Literature:**

Although this paper proposes A-PSRO as an extension of Policy Space Response Oracles (PSRO), the introduction of the Advantage function as a new metric for evaluating strategy improvement appears to be a novel contribution.

**Theoretical Claims:**

Yes, the authors provide proofs of their theoretical results in the Appendix. I checked the first several proofs, and they all appear to be correct.

---

> ### Author Rebuttal · Authors · 2025-03-28
>
> Thank you for reviewing our paper and providing valuable feedback. Our responses are as follows. We hope these responses address your concerns and that you will consider raising the score of this paper.
>
> Regarding the computational complexity of the LookAhead module, we will explain it from both theoretical and experimental perspectives.
>
> For experimental verification, please refer to Figure 9 on the last page of the appendix in the main paper. Here, we provide a detailed explanation of this figure. In our experiments, the time-consuming modules include meta-game solving, diversity-based strategy exploration, and non-diversity-based strategy exploration. Among them, the experimental code only differs in the last module between A-PSRO and other algorithms.
>
> From Figure 9b and empirical analysis, it can be observed that the solving time of the meta-game with fictitious play is an exponential function of the population size. A-PSRO has the longest runtime, indicating that A-PSRO has the largest population size during training (additional experiments show that after 200 iterations, the population size of A-PSRO is more than twice that of other algorithms). Considering that in the pipeline improvement, the PSRO algorithm does not expand the population at every iteration but only adds new strategies when the existing ones converge (see Algorithm 2 for details), this demonstrates that A-PSRO's strategy exploration quickly improves the existing strategies in the population to optimal.
>
> From Figure 9a, we can see that if only the LookAhead module is used (ours without diversity), the time spent on strategy exploration in A-PSRO increases almost linearly. From other algorithms (which perform diversity exploration with a certain probability), it can be observed that diversity exploration leads to a nonlinear increase in the time per iteration. This suggests that using the advantage function as an evaluation metric does not introduce more computational complexity compared to diversity metrics.
>
> Next, we provide a theoretical explanation. Assume that payoff U is an [n, n] matrix, and population $P_i$ and $P_j$ are [p, n] matrixs. The current meta-equilibrium $\pi$ is an [n, 1] vector, and the update step size is d.
>
> Taking the classic EC diversity metric in Equation (38) as an example:
>
> $\operatorname{EC}(\mathcal{P}_i \mid \mathcal{P}_j) := \operatorname{Tr} (\mathbf{I}-(\mathcal{L}+\mathbf{I})^{-1})$
>
> $\mathcal{L} = \mathcal{M}_i \mathcal{M}^T_i, \ \mathcal{M}_i =  \mathcal{P}_i \times U_i \times \mathcal{P}_j$
>
> Its computational complexity per iteration is $O(pn^2 + p^2n + p^3)$. Additionally, this process requires the exploration of every update directions in pure strategy space to get the one that maximizes diversity.Thus the actual computational complexity is $n \times O(pn^2 + p^2n + p^3) = O(pn^3 + p^2n^2 + p^3n)$.
>
> For the LookAhead process, here is the computation process in our code. First, repeat $\pi$ into an [n, n] matrix $Q$, and then the LookAhead update direction can be obtained through
>
> $min([Q \cdot (1-d)+I \cdot d] \times U \times I).argmax()$
>
> This process has a computational complexity of $O(n^3)$, which is independent of the population size, consistent with the linear time growth observed in the experiments, and lower than the complexity of diversity-based exploration.
>
> The above results are conclusions for zero-sum games. For general-sum games, we have already mentioned in the paper that exploring multiple oracles incurs higher computational costs. Simplifying this process is part of our future work.
>
> Regarding the application of A-PSRO to sequential decision-making in extensive-form games, we discuss it from the following perspectives. First, if the game allows direct extraction of (major) pure strategies and can obtain an empirical normal-form game through simulation, then the A-PSRO algorithm proposed in this paper can be directly applied. In this case, the advantage function can be computed directly and will not introduce higher computational complexity compared to diversity exploration.
>
> For the commonly used approach in PSRO, where RL is used to train the best response as a new strategy, we provide discussions in the "A-PSRO for Large-scale Games" section. For RL processes based on policy gradients, applying A-PSRO requires computing the gradient of a weighted sum of the reward $R$ and advantage $V$. In this case, the optimal response predictor mentioned in the paper is needed to simulate the possible optimal responses of opponents under different strategies $\pi$.
>
> This indeed requires a large amount of data for training. However, given that using RL to compute the best response is already time-consuming, and that introducing the advantage function can bring sublinear deterministic improvements, we believe this trade-off is acceptable.
>
> A typo appears in Theorem 4.1. Thanks for pointing it out.

---

> > ### Comment · Reviewer_ayi4 · 2025-04-04
> >
> > Thanks for the author's responses.
> >
> > I am confused by the answer about extending to extensive-form games: "This indeed requires a large amount of data for training. However, given that using RL to compute the best response is already time-consuming, and that introducing the advantage function can bring sublinear deterministic improvements, we believe this trade-off is acceptable.". Does this mean that applying the method to extensive-form games would still incur a significant time cost—although potentially less than that required for best response computation—and also introduce additional overhead for training the optimal response predictor?

---

> > > ### Author Response · Authors · 2025-04-08
> > >
> > > Thank you for your response. We would like to provide a detailed explanation of this issue, in the hope of alleviating your concerns about our paper. We hope you will consider raising your score.
> > >
> > > First, as we have already argued in our previous response, A-PSRO has a lower time complexity in strategy exploration compared to diversity-based algorithms. This result is also formally proven in the appendix at the end of the main text. This indicates that from the perspective of strategy exploration alone, A-PSRO does not incur greater computational complexity.
> > >
> > > Below, we focus on explaining the computational complexity of the best response predictor.
> > >
> > > To address this issue, and to demonstrate that A-PSRO is not only efficient in solving normal-form games but also applicable to extensive-form games, we conducted experiments in the widely-used Leduc Poker environment.
> > >
> > > We separately recorded the time consumption of the best response predictor, RL-based strategy exploration, and other components. As mentioned in the section "A-PSRO for Solving Large-Scale Games" in our paper, the improvement to strategy exploration brought by the best response predictor does not require 100% accuracy. Therefore, we trained the predictor using varying amounts of data.
> > >
> > > The results show that even with the smallest training dataset, the exploitability is comparable to or slightly better than that of the standard PSRO framework. From the perspective of time consumption, the most time-consuming component in the PSRO framework for Leduc Poker is the RL module. A-PSRO computes the advantage function via neural networks, which introduces negligible additional cost during the strategy exploration phase, making it nearly indistinguishable from standard PSRO in this regard.
> > >
> > > In contrast, the time spent training the best response predictor in A-PSRO is significantly less than the total time spent on the RL component. Moreover, under the same number of iterations, the exploitability of A-PSRO is 10%–20% lower than that of other PSRO algorithms.
> > >
> > > Overall, even when taking into account the training cost of the best response predictor, A-PSRO achieves lower exploitability within the same time consumption.
> > >
> > > Furthermore, as mentioned in our previous response, the strategy exploration process of A-PSRO does not rely on population size, unlike diversity-based exploration methods, and it has a lower theoretical computational complexity. Therefore, we believe that introducing the advantage function does not incur additional computation, while it improves the equilibrium-solving process.
> > >
> > > We plan to add Leduc Poker's experiments to the main paper and hope this addresses your concerns.

---

### Official Review · Reviewer_ttsB · 2025-03-14

**Overall Recommendation:** 4

**Summary:**

This paper defines “Advantage” in 0s games and 2p simplified games as the value of a policy can achieve given all other policies in the strategy profile are playing as their best response. The authors thus derive A-PSRO with Diversity and LookAhead for large-scale games. The authors thus proposed A-PSRO based on the advantage defined and shows the algorithm retains sublinear convergence rate in 0s games. Experiments show that the proposed A-PSRO with LookAhead can achieve better exploitability across domains compared with baselines.

**Claims And Evidence:**

Claim 1: introduced A-PSRO as an improved equilibrium solver for empirical games

Yes there are supportive evidence by experiments

Claim 2: The paper studies the theoretical properties of the proposed advantage functions

Yes these are supported by the theoretical analysis

**Essential References Not Discussed:**

N/A

**Experimental Designs Or Analyses:**

This paper conducted experiments in several matrix games, yet it is still to be examined in more complex games with extended form. Moreover, the scalability is questionable as the calculation of the advantage function requires an approximation of the BR and the computation complexity is not specifically discussed.

**Methods And Evaluation Criteria:**

- The paper uses exploitability and reward for 2p0s and general-sum games in a few matrix games which makes sense to me.
- The paper does not explicitly present how the advantage function, diversity and look ahead are used within the PSRO algorithm in the main text. I suggest you put an algorithm block either in the main text or appendix.

**Other Comments Or Suggestions:**

N/A

**Other Strengths And Weaknesses:**

- Strength
    - Clear demonstrations of the game solution dynamics and visualization of the advantage functions in the experimental results
- Weakness
    - The authors focus too much on the theory instead of the a clear presentation of the method. I was very distracted by the theorems when I tried to understand the methodology. I suggest that you present in the following way:
        - Here is PSRO
        - Here is how we utilize advantage function
        - Here is the Look ahead module
        - Here is the diversity module
        - then go over the theorems under different game properties
    - state clearly about the limitation of your method (e.g. only applies to xxx games)

**Questions For Authors:**

Q1: Can you compare the computational cost between the A-PSRO and PSRO for a single iteration?

Q2: The exploitability of PSD-PSRO, although in different games, is pretty low (at a scale of 10^-1) in the paper [3], while it is at a scale of 10^0 in this paper (Figure 6a). Can you elaborate on the potential reasons for the phenomenon and do the experiment domains of that paper can apply to your work?

[3]Yao, J., Liu, W., Fu, H., Yang, Y., McAleer, S., Fu, Q., & Yang, W. (2023). Policy space diversity for non-transitive games. *Advances in Neural Information Processing Systems*, *36*, 67771-67793.

**Relation To Broader Scientific Literature:**

- I think the paper has already discussed enough related works regarding the PSRO line of work in the paper.
- I wonder how the advantage is related to the counterfactual regret minimization (e.g. [1-2]) lines of work.

[1] Zinkevich, M., Johanson, M., Bowling, M., & Piccione, C. (2007). Regret minimization in games with incomplete information. *Advances in neural information processing systems*, *20*.

[2] Brown, N., Lerer, A., Gross, S., & Sandholm, T. (2019, May). Deep counterfactual regret minimization. In *International conference on machine learning* (pp. 793-802). PMLR.

**Theoretical Claims:**

- The paper studies the convergence rate with A-PSRO and introduces some theoretical properties of the advantage function
    - I think the results makes sense to me, but I did not fully check the correctness of all the proofs.

---

> ### Author Rebuttal · Authors · 2025-03-28
>
> Thank you for reviewing our paper and providing valuable feedback. We appreciate your recognition of our work. Our responses and modifications are as follows. We hope these responses address your concerns.
>
> Due to page limitations in the main text, we have placed the algorithmic details of A-PSRO in the appendix. If the final version allows for additional pages, we will move the main algorithm into the main text. Thank you for your suggestion.
>
> Regarding the computational complexity and scalability of A-PSRO, we have addressed this issue in our response to other reviewers. In general, A-PSRO has lower computational complexity compared to diversity-based exploration methods.
>
> Regarding the relationship between A-PSRO and the CFR algorithm, we have also considered this question. As far as we know, CFR is mainly applied to imperfect information games. In fact, how to compute the advantage in imperfect information games is an issue we are currently considering.
>
> For imperfect information games, our idea is to approximate the advantage of different strategies for each information set. For a given strategy, we can use Monte Carlo simulations to obtain rewards under different samplings and opponent strategies. Then, for each sample, we select the strongest opponent and apply a weighting to simulate the advantage function. The main challenge is that this leads to a large computational complexity, and we are considering using methods such as Transformer to improve efficiency.
>
> Regarding the comparison with the PSD-PSRO algorithm in Figure 6, it is worth noting that this experiment was conducted in an environment with three agents. PSD-PSRO, as mentioned in its notation section, is primarily designed for two-player zero-sum games and has not been optimized for multi-agent scenarios. As a result, it shows higher exploitability in this case. For A-PSRO, we designed a method to approximate the advantage function in multi-agent systems, resulting in better performance.
>
> Thank you for your suggestion. We will provide a detailed explanation of the theoretical properties of the algorithm and how A-PSRO applies in different scenarios in separate sections of the main text. We will also add clarifications on in which scenarios A-PSRO is effective, where it may lead to higher computational complexity, and where its effectiveness remains uncertain.
>
> We appreciate your recognition of our work, and we hope the above responses address your concerns. We hope you continue to support our paper.

---

### Official Review · Reviewer_vFFK · 2025-03-14

**Overall Recommendation:** 3

**Summary:**

The authors propose an extension of PSRO to normal-form games with large-scale action spaces. They incorporate an advantage function to guide strategy exploration and speed up convergence to NE and improve joint rewards in general-sum normal-form games.

**Claims And Evidence:**

- The authors claim to establish an equivalence between advantage maximization and Nash equilibrium. Theoretically, this seems to follow.
- The authors claim that including advantage maximization in normal-form PSRO allows their method to achieve higher joint rewards in general-sum games.

They empirically support these claims by showing that they outperform other PSRO variants in normal form games.


A-PSRO is positioned as a solver for large-scale normal-form games, yet the paper does not compare it to standard methods for solving such games, such as linear programming, fictitious play, or regret minimization. Furthermore, PSRO and its extensions are predominantly designed for and applied to extensive-form and partially-observable Markov games, but the paper does not address how A-PSRO relates to or extends beyond these settings. This omission raises concerns about the generality and significance of the proposed approach.

**Essential References Not Discussed:**

The authors focus on solving normal-form games, and should include foundational algorithms that are still used today to solve them [1,2,3]

[1] Von Neumann's minimax theorem: v. Neumann, J. "Zur theorie der gesellschaftsspiele." Mathematische annalen 100.1 (1928): 295-320.

[2] Lemke-Howson Algorithm: Lemke, Carlton E., and Joseph T. Howson, Jr. "Equilibrium points of bimatrix games." Journal of the Society for industrial and Applied Mathematics 12.2 (1964): 413-423.

[3] Multiplicative Weights Update Algorithm: Freund, Yoav, and Robert E. Schapire. "Adaptive game playing using multiplicative weights." Games and Economic Behavior 29.1-2 (1999): 79-103.

**Ethical Review Concerns:**

I have no ethical concerns regarding this paper.

**Experimental Designs Or Analyses:**

The experimental design (exploitability/joint reward vs training iterations) is sound.

**Methods And Evaluation Criteria:**

The proposed evaluation criteria does not make sense for the problem discussed in the paper. A-PSRO is designed as a large-scale normal-form game solver, yet it is not compared with normal-form game algorithms other than PSRO variants.

**Other Comments Or Suggestions:**

Error bounds need to be added in Figure 2 and Figure 7.

**Other Strengths And Weaknesses:**

Strengths:
- The advantage metric provides a useful signal to speed up convergence of PSRO in normal-form games.

Weaknesses:
- It is unclear how useful this method is for normal-form games without comparing to non-PSRO methods.

**Questions For Authors:**

1) What constitutes a large-scale normal-form game in this setting? At what scale would A-PSRO be more preferable than traditional normal-form game algorithms like linear programming, fictitious play, and regret minimization? Why is a PSRO-style approach necessary in this context compared to other methods?

2) Why evaluate against Pipeline-PSRO? It speeds up wall-time performance in extensive-form games by concurrently pretraining behavioral deep RL best-responses. It's applicability as a normal-form game solver is extremely limited.

3) This question generally extends to comparing against most of the other PSRO baselines. If you are trying to solve large-scale normal-form games quickly, why not compare to other methods designed to solve normal form games, like linear programming approaches, fictitious play, or regret matching?

4) Is your goal to extend A-PSRO to extensive-form games where PSRO is generally applied? If so, this motivation is very unclear, and the non-trivial challenges in extending the advantage metric to extensive-form games need to be discussed.

**Relation To Broader Scientific Literature:**

The authors should be clear early on that their decision to consider PSRO as a solution to normal-form games is unusual. They should provide better context that PSRO and nearly all of its extensions are designed to solve games with sequential decision making like extensive-form and partially-observable Markov games. PSRO is typically considered as a extension of the normal Double Oracle algorithm to sequential interaction games.

It also needs to be made more clear that most of the works cited in this paper address games with sequential interaction, not normal-form games (except as sanity checks and stepping stones to other game representations).

**Theoretical Claims:**

I did not rigorously check the proofs

Theorem 4.8 suggests that training a best-response approximator with sufficient accuracy for a game will guarantee a sublinear convergence rate in symmetric zero-sum normal-form games. It needs to be discussed that the amount of time necessary to calculate a sufficient training dataset of many best response targets could be large.

---

> ### Author Rebuttal · Authors · 2025-03-28
>
> Thank you for reviewing our paper and providing valuable feedback. Our responses and modifications are as follows. We hope these responses address your concerns and that you will consider raising the score of this paper.
>
> We would like to emphasize that the motivation of this paper is the improvement of the strategy exploration process in the PSRO algorithm. Discussing this problem in the context of normal-form games is mainly because the properties of the advantage function and the convergence of the algorithm can be identified. We mentioned in our response to Reviewer DSTe that we were conducting experiments with Leduc Poker and plan to add the results to the main text.
>
> The use of normal-form games in the title is primarily for the following reasons.
>
> First, previous PSRO variants have typically improved the solution in zero-sum games, while we aim to demonstrate that A-PSRO can also improve the solution efficiency for general-sum games.
>
> Additionally, the main reason for considering normal-form games comes from the paper "Real World Games Look Like Spinning Tops" [1]. For any extensive-form game, extracting all pure strategies can define a corresponding normal-form game (especially in fully observable scenarios). Further discussions in [1] indicate that pure strategies with a wide range of skills extracted from large-scale extensive-form games (such as StarCraft) can also define a normal-form game. This can be viewed as an empirical game containing the most frequent strategies in the original game. The strategies obtained by solving the empirical game are also important for many problems. There are several works about empirical games:
>
> "Choosing samples to compute the heuristic strategy Nash equilibrium"
>
> The primary experimental environment in this paper is based on [1], with some of the experiments coming from both complete extensive-form games and their simplified forms (such as AlphaStar, Go, and Kuhn Poker). In the paper [1], it is mentioned that applying population-based policy learning in these environments is the most effective, so we mainly compare the PSRO algorithm.
>
> This experimental design is identical to previous diversity-based PSRO variants (such as DPP-PSRO and UDF-PSRO). Therefore, we believe the use of normal-form games is justified, as it emphasizes the inclusion of both zero-sum and general-sum games.
>
> One of the contributions of this paper is that A-PSRO improves the reward of equilibrium in general-sum games. This is one reason why it is more preferable than traditional methods.
>
> Regarding the comparison with non-PSRO algorithms, traditional algorithms generally perform inefficiently in the game environments presented in [1]. For example, the previous work "Open-ended Learning in Symmetric Zero-sum Games" compared Self-Play, and "Pipeline PSRO: A Scalable Approach for Finding Approximate Nash Equilibria in Large Games" compared Fictitious Play (regret minimization is primarily used for solving imperfect information games and will not be discussed here). Given that A-PSRO is almost identical to the PSRO framework except for strategy exploration, and that PSRO-based algorithms perform the best in the experimental environments of this paper, we believe this comparison is reasonable.
>
> Since linear programming is not a learning-based approach to solving equilibria, we here mainly discuss Fictitious Play. As mentioned in our response to reviewer ayi4, the time spent on strategy exploration in the PSRO framework is small compared to the time spent on meta-game solving. Compared to running Fictitious Play directly in the original game, efficient exploration can solve equilibria in small populations. We found experimentally that the population size only needs to be less than 10% of the original game.Considering that the solution of the meta-game is of approximately exponential complexity, this process greatly improves efficiency.
>
> On the populations obtained from A-PSRO exploration, Fictitious Play only requires about $10^3$ of iterations to reach $10^{-4}$ exploitability.In contrast, it takes about $10^4$ iterations or more directly using fictitious play.
>
> Considering that none of the other PSRO algorithms compared to traditional methods in the environment used in this paper, we believe it is reasonable. We could add these comparisons, but we think they may perform inefficiently.
>
> As for why we compare with Pipeline-PSRO, the idea of using multiple learners in Algorithm 2 has been applied in both DPP-PSRO and UDF-PSRO, significantly improving performance. Our experimental design and baselines are almost identical to those in these two papers, and this paper also uses multiple learners to update strategies simultaneously.
>
> About the esential references not discussed, thank you for pointing this out, and we will add the missing references. We will also add error bounds in the two figures.
>
> [1] Czarnecki et al. Real World Games Look Like Spinning Tops, NeurIPS 2020.

---

> > ### Comment · Reviewer_vFFK · 2025-04-09
> >
> > Thank you for your response.
> >
> > To me, the argument that any extensive-form game can be converted into an (exponentially large) normal-form game is not a good reason that normal-form games should be primarily considered. If we consider a large extensive-form game, the corresponding normal-form game would be intractable for any method. If we consider a restricted meta-game of useful strategies, typically this is a small part of an algorithm like PSRO (applied to extensive-form games) that does not take a significant portion of the running time.
> >
> > I would be happy to consider extensive-form results with Leduc, but currently, only a single intermediate exploitability data point has been given in response to reviewer DSTe. Without an exploitability curve, we can't deduce any actual comparison from this, as one method could overtake the other. I am otherwise unconvinced that this method necessarily scales well in extensive-form.
> >
> > I am unconvinced that a wall-time comparison to linear programming should not be done. It should be demonstrated at what game size is learning even necessary here.
> >
> > Concerning Pipeline-PSRO, if P-PSRO and A-PSRO enjoy benefits from employing multiple learners simultaneously, that makes an unfair x-axis In Figures 2, 4, 6, 7. Graphs using "Training Iterations" as the x-axis make any method with multiple concurrent learners look better than it is compared to single learner methods like PSRO. Wall-time or total learning updates (number of learners * Training Iterations) would be a fair x-axis.
> >
> > I maintain my current score.

---

> > > ### Author Response · Authors · 2025-04-09
> > >
> > > Thank you for your response.We believe that these problems are not flaws in the paper itself, but that there may be some misunderstandings.We hope that the following responses will address your questions.
> > >
> > > Regarding normal-form games, we would like to emphasize that the main focus of this paper is on improving the PSRO framework, rather than being limited to solving normal-form games specifically. For the proposed advantage function, it can be computed exactly in normal-form games, while in extensive-form games it requires approximation.
> > >
> > > We have proven through multiple theorems that, under the assumption of exact advantage computation, the convergence rate to equilibrium can be improved to sublinear. Even with approximate computation, this result can still be achieved within a certain error bound. Moreover, the introduction of the advantage function enables convergence to equilibria with higher rewards in general-sum games.
> > >
> > > Given that the definition of normal-form games itself encompasses all games, this indicates that our theory applies to all fully observable games. We believe that, for a theoretical contribution, this is both meaningful and significant.
> > >
> > > Regarding the experimental results on Leduc Poker, we refrained from including specific plots due to anonymity and possible violations concerns. In experiment, all other modules in the experiment were kept exactly the same as those in PSD-PSRO (SOTA), with the only difference being in the strategy exploration process. From the results, A-PSRO achieved a faster decrease in exploitability during the early stages. As the number of iterations increased, the effect of the advantage function diminished to some extent, but A-PSRO still outperformed PSD-PSRO in the final results. Below are the average exploitability values at different stages of the iterations.
> > >
> > > Episodes(1e4)       $\quad$  10       $\quad$     $\quad$       50                      $\quad$        $\quad$                 100             $\quad$                 $\quad$                 200
> > >
> > > PSD-PSRO   $\quad$ $1.2 \times 10^0$  $7.3 \times 10^{-1}$  $5.2 \times 10^{-1}$  $3.9 \times 10^{-1}$
> > >
> > > A-PSRO    $\quad$ $\quad$ $0.8 \times 10^0$  $5.0 \times 10^{-1}$ $3.3 \times 10^{-1}$  $2.7 \times 10^{-1}$
> > >
> > > Regarding comparisons with linear programming, we believe the key difference lies in that learning-based methods approximate equilibria rather than solving them exactly. We believe that these two methods are not directly comparable, as linear programming requires significantly more time to compute an exact solution, whereas learning-based algorithms can obtain an approximate solution in much less time.
> > >
> > > If we consider an exploitability level of $10^{-2}$ to be sufficiently close to equilibrium, then from the final page of our appendix, it can be observed that A-PSRO requires fewer than 50 iterations to achieve this, with a total runtime less than 1 minute. However, when using linear programming, the number of variables and constraints involved can be on the order of $10^3–10^4$ (e.g., AlphaStar, Simplified Go game). To the best of our knowledge, even tools like Gurobi are unlikely to achieve an exact solution within 1 minutes. It may take one or more hours to complete the solution. Therefore, we argue that learning-based methods like A-PSRO have clear advantages in terms of time efficiency.
> > >
> > > Regarding Pipeline-PSRO, we have reviewed all the methods we compared in the paper as well as their corresponding implementations. Except for standard PSRO, all other compared methods used the pipeline improvement in their codebases, including P-PSRO, DPP-PSRO, UDF-PSRO, and PSD-PSRO. We reproduced the code for all of these methods, and without exception, they use multiple learners to update strategies (i.e., the pipeline improvement), and plot performance against "Training Iterations" on the x-axis. We believe this is a standard comparasion and does not introduce any unfairness.
> > >
> > > We would like to reemphasize that, during experiments, A-PSRO differs from other methods only in the strategy exploration module. In result evaluation, meta-strategy solving, and plotting, A-PSRO is implemented identically to the baselines. Considering that the compared methods are all PSRO variants, we believe this setup is appropriate and fair.
> > >
> > > In summary, we have already shown that compared to traditional linear programming and fictitious play, A-PSRO achieves solutions with significantly less computational time. This demonstrates that even when considering only normal-form games, A-PSRO still delivers the best overall performance. In addition, we are conducting experiments to further validate its applicability in extensive-form games, and the current results have already shown that A-PSRO can work effectively in such settings.
> > >
> > > We hope the above response addresses your concerns, and we sincerely hope that you will consider raising the score for our submission.

---

### Official Review · Reviewer_DSTe · 2025-03-14

**Overall Recommendation:** 3

**Summary:**

The paper addresses the challenge of solving Nash equilibria in normal-form games, particularly for games with large strategy spaces. Traditional PSROs and their variants have been effective in learning equilibria but often lack an efficient metric to evaluate and guide strategy improvement. This limitation affects their convergence efficiency and performance in both zero-sum and general-sum games.

As a solution, the proposed method leverages the Advantage function—a new evaluative metric for guiding strategy updates toward Nash equilibria. Theoretical analysis shows that the Advantage function has desirable properties like convexity and Lipschitz continuity, ensuring more efficient equilibrium approximation. Furthermore, by integrating a LookAhead module for refining strategy updates and supports neuralization, APSRO makes it scalable to large games. Experiments in zero-sum and general-sum games demonstrate that A-PSRO significantly reduces exploitability, finds higher-reward equilibria, and outperforms existing PSRO variants in convergence efficiency and reward maximization.

**Claims And Evidence:**

Overall, most claims in the paper are supported by theoretical analysis and empirical results, but a few areas could benefit from further clarification or stronger evidence, I list them as follows:

> A-PSRO has a deterministic convergence rate advantage over diversity-based PSRO methods.

The paper claims that advantage-based exploration leads to more deterministic convergence, but experimental evidence is limited. The convergence rate is not explicitly analyzed against diversity-based PSRO variants. A more detailed convergence speed comparison would strengthen this claim.

> The Advantage function helps overcome the limitations of diversity-based exploration in PSRO.

While A-PSRO does reduce exploitability, the role of diversity exploration in complementing the Advantage function is not deeply analyzed. The interaction between diversity-based methods and the Advantage function needs more empirical justification, as some results (e.g., in Transitive games) suggest diversity alone may sometimes perform better.

**Essential References Not Discussed:**

Relevant Missing Work:

1. Neural Population Learning Beyond Symmetric Zero-Sum Games (Liu et al., 2024)
2. Pipeline-PSRO: A Scalable Approach for Finding Approximate Nash Equilibria in Large Games (McAleer et al., 2020)

**Experimental Designs Or Analyses:**

For the ablation test for:
- A-PSRO without LookAhead (LA)
- A-PSRO without Diversity

The effect of LookAhead is clear, as removing it worsens exploitability reduction. But a potential issue is: the interaction between Advantage-based updates and Diversity is not fully explored.

**Methods And Evaluation Criteria:**

The proposed methods and evaluation criteria make sense for the problem of solving Nash equilibria in normal-form games. However, there are a few areas where the evaluation could be expanded or clarified, e.g.

> Computational Efficiency Analysis Missing

The paper does not provide a runtime or scalability comparison against existing PSRO methods. Since Advantage computation and LookAhead introduce additional complexity, an evaluation of training time would help assess practical feasibility.

> Limited Real-World Validation

The experiments focus on synthetic normal-form games, which are common in game theory but may not directly translate to real-world multi-agent learning problems (e.g., poker, RTS games like StarCraft). Testing A-PSRO in a multi-agent reinforcement learning (MARL) setting would demonstrate its broader applicability.

**Other Comments Or Suggestions:**

N/A

**Other Strengths And Weaknesses:**

Has been discussed in previous questions

**Questions For Authors:**

N/A

**Relation To Broader Scientific Literature:**

A-PSRO is strongly connected to prior work in PSRO methods, Nash equilibrium learning, and multi-agent learning. It extends these areas by introducing the Advantage function, which provides a principled optimization target for reducing exploitability and selecting better equilibria. While diversity-based PSRO methods rely on heuristic exploration, A-PSRO offers a mathematically grounded alternative, making it a meaningful contribution to equilibrium learning research.

**Theoretical Claims:**

No obvious errors were found

---

> ### Author Rebuttal · Authors · 2025-03-28
>
> Thank you for reviewing our paper and providing valuable feedback. Our responses and modifications are as follows. We hope these responses address your concerns and that you will consider raising the score of this paper.
>
> Regarding the deterministic convergence rate, the explanation is as follows. The convergence of the PSRO algorithm depends on both the strategy exploration process and the meta-game solving process. Our improvement only addresses the strategy exploration process, and we have proven its convergence properties.
>
> We believe you might have misinterpreted the figure in the paper. For Transitive games, the best-performing algorithm is A-PSRO without diversity (not with diversity), which indicates that the algorithm using only LookAhead performs the best.
>
> In fact, we analyze the effects of both in the main paper and the supplementary materials. For games that mainly exhibit transitivity, the LookAhead module can converge very quickly to the equilibrium without relying on diversity exploration. This can be viewed as an optimisation process in a convex function with a significant gradient, where rapid convergence can be achieved by directly using the gradient pointing to a local maximum. Games with strong cyclic dimensions can be viewed as convex functions without significant gradient. Although the LookAhead module can converge to the equilibrium, the convergence may be slow in the early stage. Diversity exploration, which is akin to a stochastic optimization process, may directly update the strategy to a region very close to the equilibrium. Experimental results also show that the combination of both methods yields the best performance.
>
> Regarding the computational complexity of A-PSRO, please refer to our response to Reviewer ayi4.
>
> Regarding the code, the explanation is as follows. Due to insufficient time for code refinement, there may be some redundancy issues. We will make the necessary modifications in the future.
>
> About the esential references not discussed, thank you for pointing this out, and we will add the missing references.
>
> Regarding experiments on A-PSRO in more real-world games, we provide the following explanation. Our initial experiments primarily referenced DPP-PSRO and UDF-PSRO. These two works only conducted experiments on the normal-form games as presented in ”Real World Games Look Like Spinning Tops”. These experimental environments consist mainly of normal-form games consisting of pure strategies extracted from extensive-form games, which serve as empirical games that can model the interactions of the agents in the game. In fact, these experimental environments include many real-world extensive-form games (such as Kuhn Poker, Starcraft, etc.).
>
> Given that reviewers have raised concerns about this, we recently conducted experiments in Leduc Poker, which is commonly used for evaluating PSRO algorithms, following the approach of PSD-PSRO. The application of A-PSRO in this game primarily requires approximating the computation of the advantage function, as detailed in the A-PSRO for Large Scale Games section.
>
> The current experimental results of exploitability are:
>
> PSD-PSRO: $4 \times 10^{-1}$, A-PSRO: $3 \times 10^{-1}$.
>
> Due to time constraints, A-PSRO has not been fully fine-tuned. We believe that further improvements in the code could yield better results. If this comparison is necessary, we will add it to the main paper.

---

### Decision · Program_Chairs · 2025-05-01

**Decision:**

Accept (poster)

**Comment:**

In contrast to the standard best-response strategy expansion of PSRO, the authors suggest proposing new strategies that optimize the “advantage”. The reviewers generally appreciated this novel approach and the accompanying guarantees. A-PSRO is shown to achieve lower levels of exploitability and higher joint reward in several NFGs. In response to the reviewers, the authors added results for the EFG Leduc poker. This remains a concern as PSRO is primarily recognized as a technique to extend NE-approximation to games so large that traditional techniques cannot cope.